# Independent evolution of ancestral and novel defenses in a genus of toxic plants (*Erysimum*, Brassicaceae)

Tobias Züst[1]*, Susan R Strickler[2], Adrian F Powell[2], Makenzie E Mabry[3], Hong An[3], Mahdieh Mirzaei[2], Thomas York[2], Cynthia K Holland[2†], Pavan Kumar[2‡], Matthias Erb[1], Georg Petschenka[4], José-María Gómez[5], Francisco Perfectti[6], Caroline Müller[7], J Chris Pires[3], Lukas A Mueller[2], Georg Jander[2]

[1]Institute of Plant Sciences, University of Bern, Bern, Switzerland; [2]Boyce Thompson Institute, Ithaca, United States; [3]Division of Biological Sciences, University of Missouri, Columbia, United States; [4]Institut für Insektenbiotechnologie, Justus-Liebig-Universität Giessen, Giessen, Germany; [5]Department of Functional and Evolutionary Ecology, Estación Experimental de Zonas Áridas (EEZA-CSIC), Almería, Spain; [6]Research Unit Modeling Nature, Department of Genetics, University of Granada, Granada, Spain; [7]Department of Chemical Ecology, Bielefeld University, Bielefeld, Germany

*For correspondence:
tobias.zuest@ips.unibe.ch

Present address: †Thompson Biology Lab, Williams College, Williamstown, United States; ‡Corteva Agriscience, Des Moines, United States

Competing interests: The authors declare that no competing interests exist.

**Abstract** Phytochemical diversity is thought to result from coevolutionary cycles as specialization in herbivores imposes diversifying selection on plant chemical defenses. Plants in the speciose genus *Erysimum* (Brassicaceae) produce both ancestral glucosinolates and evolutionarily novel cardenolides as defenses. Here we test macroevolutionary hypotheses on co-expression, co-regulation, and diversification of these potentially redundant defenses across this genus. We sequenced and assembled the genome of *E. cheiranthoides* and foliar transcriptomes of 47 additional *Erysimum* species to construct a phylogeny from 9868 orthologous genes, revealing several geographic clades but also high levels of gene discordance. Concentrations, inducibility, and diversity of the two defenses varied independently among species, with no evidence for trade-offs. Closely related, geographically co-occurring species shared similar cardenolide traits, but not glucosinolate traits, likely as a result of specific selective pressures acting on each defense. Ancestral and novel chemical defenses in *Erysimum* thus appear to provide complementary rather than redundant functions.

## Introduction

Plant chemical defenses play a central role in the coevolutionary arms race with herbivorous insects. In response to diverse environmental challenges, plants have evolved a plethora of structurally diverse organic compounds with repellent, antinutritive, or toxic properties (*Fraenkel, 1959*; *Mithöfer and Boland, 2012*). Chemical defenses can impose barriers to consumption by herbivores, but in parallel may favor the evolution of specialized herbivores that can tolerate or disable these defenses (*Cornell and Hawkins, 2003*). Chemical diversity is likely evolving in response to a multitude of plant-herbivore interactions (*Salazar et al., 2018*), and community-level phytochemical diversity may be a key driver of niche segregation and insect community dynamics (*Richards et al., 2015*; *Sedio et al., 2017*).

For individual plants, the production of diverse mixtures of chemicals is often considered advantageous (*Romeo, 1996*; *Firn and Jones, 2003*; *Gershenzon et al., 2012*; *Forbey et al., 2013*; *Richards et al., 2016*). For example, different chemicals may target distinct herbivores (*Iason et al.,*

**eLife digest** Plants are often attacked by insects and other herbivores. As a result, they have evolved to defend themselves by producing many different chemicals that are toxic to these pests. As producing each chemical costs energy, individual plants often only produce one type of chemical that is targeted towards their main herbivore. Related species of plants often use the same type of chemical defense so, if a particular herbivore gains the ability to cope with this chemical, it may rapidly become an important pest for the whole plant family.

To escape this threat, some plants have gained the ability to produce more than one type of chemical defense. Wallflowers, for example, are a group of plants in the mustard family that produce two types of toxic chemicals: mustard oils, which are common in most plants in this family; and cardenolides, which are an innovation of the wallflowers, and which are otherwise found only in distantly related plants such as foxglove and milkweed. The combination of these two chemical defenses within the same plant may have allowed the wallflowers to escape attacks from their main herbivores and may explain why the number of wallflower species rapidly increased within the last two million years.

Züst et al. have now studied the diversity of mustard oils and cardenolides present in many different species of wallflower. This analysis revealed that almost all of the tested wallflower species produced high amounts of both chemical defenses, while only one species lacked the ability to produce cardenolides. The levels of mustard oils had no relation to the levels of cardenolides in the tested species, which suggests that the regulation of these two defenses is not linked. Furthermore, Züst et al. found that closely related wallflower species produced more similar cardenolides, but less similar mustard oils, to each other. This suggests that mustard oils and cardenolides have evolved independently in wallflowers and have distinct roles in the defense against different herbivores.

The evolution of insect resistance to pesticides and other toxins is an important concern for agriculture. Applying multiple toxins to crops at the same time is an important strategy to slow the evolution of resistance in the pests. The findings of Züst et al. describe a system in which plants have naturally evolved an equivalent strategy to escape their main herbivores. Understanding how plants produce multiple chemical defenses, and the costs involved, may help efforts to breed crop species that are more resistant to herbivores and require fewer applications of pesticides.

---

*2011*; *Richards et al., 2015*), or may act synergistically to increase overall toxicity of a plant (*Steppuhn and Baldwin, 2007*). However, metabolic constraints can limit the extent of phytochemical diversity within individual plants (*Firn and Jones, 2003*). Most defensive metabolites originate from a small group of precursor compounds and conserved biosynthetic pathways, which are modified in a hierarchical process into diverse, species-specific end products (*Moore et al., 2014*). As constraints are likely strongest for the early stages of these pathways, related plant species commonly share the same functional 'classes' of defensive chemicals (*Wink, 2003*), but vary considerably in the number of compounds within each class (*Fahey et al., 2001*; *Rasmann and Agrawal, 2011*).

Functional conservatism in defensive chemicals among related plants should facilitate host expansion and the evolution of tolerance in herbivores (*Cornell and Hawkins, 2003*), as specialized resistance mechanisms against one type of compound are more likely to be effective against structurally similar than structurally dissimilar compounds. This may result in a seemingly paradoxical scenario, wherein well-defended plants are nonetheless attacked by a diverse community of specialized herbivores (*Agrawal, 2005*; *Bidart-Bouzat and Kliebenstein, 2008*). For example, most plants in the Brassicaceae produce glucosinolates as their primary defense, which upon activation by myrosinase (thioglucoside glucohydrolase) enzymes upon leaf damage become potent repellents of many herbivores (*Fahey et al., 2001*). However, despite the potency of this defense system and the large diversity of glucosinolates produced by the Brassicaceae, several specialized herbivores have evolved strategies to overcome this defense, enabling them to consume most Brassicaceae and even to sequester glucosinolates for their own defense against predators (*Müller, 2009*; *Winde and Wittstock, 2011*).

Plants may occasionally overcome the constraints on functional diversification and gain the ability to produce new classes of defensive chemicals as a 'second line of defense' (*Feeny, 1977*). Although

this phenomenon is likely widespread across the plant kingdom, it has most commonly been reported from the well-studied Brassicaceae. In addition to producing evolutionarily ancestral gluco-sinolates, plants in this family have gained the ability to produce saponins in *Barbarea vulgaris* (*Shinoda et al., 2002*), alkaloids in *Cochlearia officinalis* (*Brock et al., 2006*), cucurbitacins in *Iberis* spp. (*Nielsen, 1978b*), alliarinoside in *Alliaria petiolata* (*Frisch and Møller, 2012*), and cardenolides in the genus *Erysimum* (*Makarevich et al., 1994*). These recently-evolved chemical defenses with modes of action distinct from glucosinolates have likely allowed the plants to escape attack from specialized, glucosinolate-adapted herbivores (*Nielsen, 1978b*; *Dimock et al., 1991*; *Haribal and Renwick, 2001*; *Shinoda et al., 2002*). Gains of novel defenses are expected to result in a release from selective pressures imposed by specialized antagonists, and thus may represent key steps in herbivore-plant coevolution that lead to rapid phylogenetic diversification (*Weber and Agrawal, 2014*).

The production of cardenolides by species in the genus *Erysimum* is one of the longest- and best-studied examples of an evolutionarily recent gain of a novel chemical defense (*Jaretzky and Wilcke, 1932*; *Nagata et al., 1957*; *Singh and Rastogi, 1970*; *Makarevich et al., 1994*). Cardenolides are a type of cardiac glycoside, which act as allosteric inhibitors of Na$^+$/K$^+$-ATPase, an essential membrane ion transporter that is expressed ubiquitously in animal cells (*Agrawal et al., 2012*). Cardiac glyco-sides are produced by plants in approximately sixty genera belonging to twelve plant families, and several cardiac glycoside-producing plants are known for their toxicity or medicinal uses (*Agrawal et al., 2012*; *Züst et al., 2018*). *Erysimum* is a species-rich genus consisting of diploid and polyploid species with diverse morphologies, growth habits, and ecological niches (*Al-Shehbaz, 1988*; *Polatschek and Snogerup, 2002*; *Al-Shehbaz, 2010*; *Gómez et al., 2015*). Of the *Erysi-mum* species evaluated to date, all produce some of the novel cardenolide defenses (*Makarevich et al., 1994*). Previous phylogenetic studies suggest a recent and rapid diversification of the genus, with most species divergence occurring within the last 2–3 million years (*Gómez et al., 2014*; *Moazzeni et al., 2014*), resulting in 150 to 350 extant species (*Polatschek and Snogerup, 2002*; *Al-Shehbaz, 2010*). The large uncertainty in species number reflects taxonomic challenges in this genus, which includes many species that readily hybridize, as well as cryptic species with near-identical morphology (*Abdelaziz et al., 2011*).

In most *Erysimum* species, cardenolides appear to have enabled an escape from at least some glucosinolate-adapted specialist herbivores. Cardenolides in *Erysimum* act as oviposition and feed-ing deterrents for different pierid butterflies (*Chew, 1975*; *Chew, 1977*; *Wiklund et al., 1978*; *Renwick et al., 1989*; *Dimock et al., 1991*), and several glucosinolate-adapted beetles (*Phaedon* spp. and *Phyllotreta* spp.) were deterred from feeding by dietary cardenolides at levels commonly found in *Erysimum* (*Nielsen, 1978a*; *Nielsen, 1978b*). Nonetheless, *Erysimum* plants are still attacked by a range of herbivores and seed predators, including some mammals and several glucosi-nolate-adapted aphids, true bugs, and lepidopteran larvae (*Gómez, 2005*; *Züst et al., 2018*). These herbivores likely rely on general detoxification mechanisms for tolerance of the novel defense, while to date there are no reports of de novo gains of specialized cardenolide resistance in *Erysimum* her-bivores. However, the gain of the novel defense may have facilitated host shifts in at least one carde-nolide-adapted herbivore: in addition to its main host *Digitalis purpurea* (Plantaginaceae), the seed-feeding bug *Horvathiolus superbus* (Lygaeinae) commonly feeds on seeds of *E. crepidifolium*, and is able to sequester cardenolides from both sources (Georg Petschenka, personal observations).

The gain of a novel chemical defense makes the genus *Erysimum* an excellent model system to study the causes and consequences of phytochemical diversification (*Züst et al., 2018*). While an increasing number of studies are beginning to describe taxon-wide patterns of chemical diversity in plants (e.g., *Richards et al., 2015*; *Sedio et al., 2017*; *Salazar et al., 2018*), the *Erysimum* system is unique in combining two classes of plant metabolites with primarily defensive function – although a broader role of glucosinolates is increasingly recognized (e.g., *Katz et al., 2015*). The system thus is ideally suited to evaluate the evolutionary consequences of co-expressing two functionally distinct but potentially redundant defenses. Here, we present a high-quality genome sequence assembly and annotation for the short-lived annual *E. cheiranthoides* as an important resource for future molecular studies in this system. Furthermore, we present a new phylogeny for 48 species con-structed from transcriptome sequences (*Figure 1*), corresponding to 10–30% of species in the genus *Erysimum*. We combine this phylogeny with a characterization of the full diversity of glucosinolates and cardenolides in leaves to evaluate macroevolutionary patterns in the evolution of phytochemical

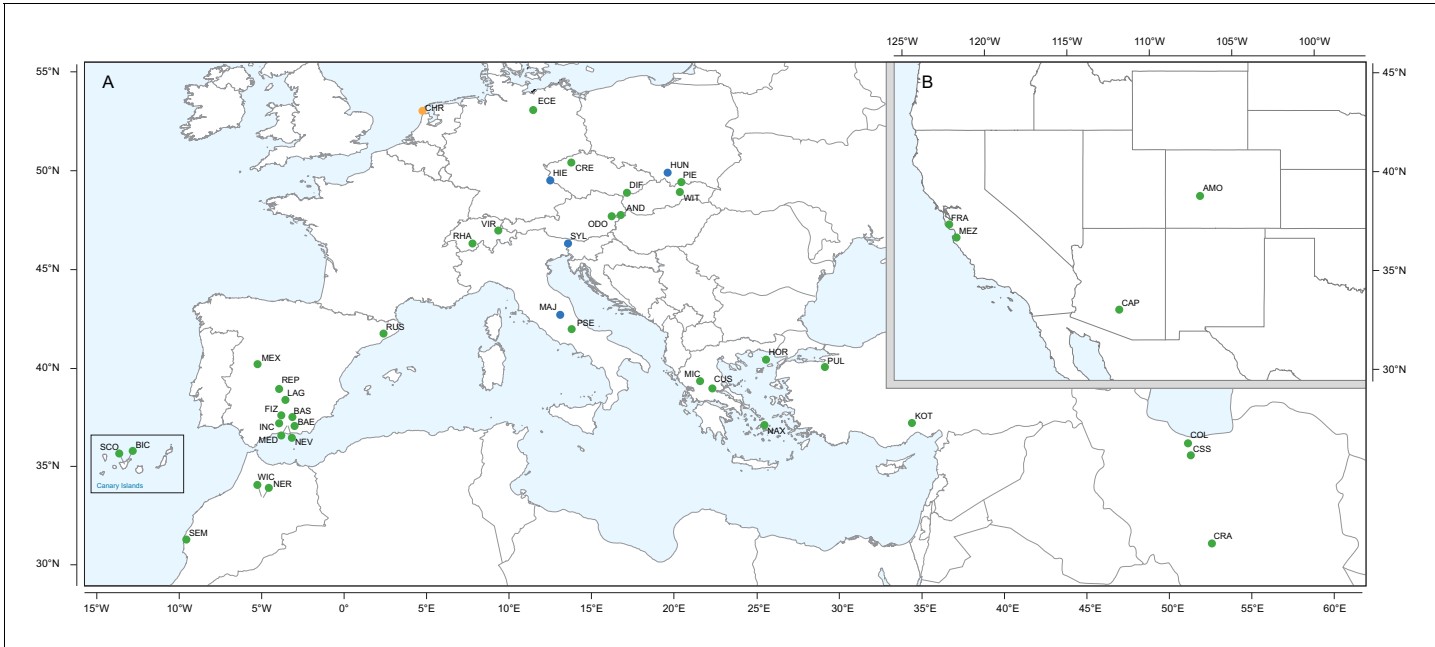

**Figure 1.** Geographic location of *Erysimum* species source populations used for transcriptome sequencing. (**A**) Source populations in Europe. Inset: The Canary Islands (28°N, 16°W) are located further westward and southward than drawn in this map. Green symbols are exact collection locations, while blue symbols indicate approximate locations based on species distributions. Seeds of the originally Mediterranean species *E. cheiri* (CHR, orange symbol) were collected from a naturalized population in the Netherlands. (**B**) Source populations in North America. Five species/accessions (ALI, ER1, ER2, ER3, ER4) could not be placed on the map due to uncertain species identity. See ***Supplementary file 1*** for more details.

diversity across the genus. We complemented the characterization of defensive phenotypes by quantifying glucosinolate-activating myrosinase activity, inhibition of animal $Na^+/K^+$-ATPase by leaf extracts, and defense inducibility in response to exogenous application of jasmonic acid (JA). By assessing co-variation of diversity, abundance and inducibility of ancestral and novel defenses, we provide evidence that these two classes of defense metabolites evolved in response to different selective pressures and appear to have specific, non-redundant functions.

## Results

### *E. cheiranthoides* genome assembly

A total of 39.5 Gb of PacBio sequences with an average read length of 10,603 bp were assembled into 1087 contigs with an N50 of 1.5 Mbp (***Table 1***). Hi-C scaffolding oriented 98.5% of the assembly into eight large scaffolds representing pseudomolecules (***Table 1***, ***Figure 2—figure supplement 1***), while 216 small contigs remained unanchored. The final assembly (v1.2) had a total length of 174.5 Mbp, representing 86% of the estimated genome size of *E. cheiranthoides* and capturing 99% of the BUSCO gene set (***Table 1***, ***Figure 2—figure supplement 2***). Sequences were deposited under Gen-Bank project ID PRJNA563696 and additionally are provided at www.erysimum.org. A total of 29,947 gene models were predicted and captured 98% of the BUSCO gene set (***Figure 2—figure supplement 3***). In the presumed centromere regions of each chromosome, genic sequences were less abundant, whereas repeat sequences were more common (***Figure 2A***). Repetitive sequences constituted approximately 29% of the genome (***Supplementary file 2***). Long terminal repeat retro-transposons (LTR-RT) made up the largest proportion of the repeats identified (***Figure 2—figure supplement 4***). Among these, repeats in the *Gypsy* superfamily constituted the largest fraction of the genome (***Supplementary file 2***). The majority of the LTR elements appeared to be relatively young, with most having estimated insertion times of less than 1 MYA (***Figure 2—figure supplement 5***). Synteny analysis showed evidence of several chromosomal fusions and fissions between the eight chromosomes of *E. cheiranthoides* and the five chromosomes of Arabidopsis (***Figure 2B***).

**Table 1.** Assembly metrics for the *E. cheiranthoides* genome: v0.9=Falcon +Arrow assembly results, v1.2=genome assembly after Hi-C scaffolding and Pilon correction.

|  | v0.9 | v1.2 pseudomolecules and contigs | v1.2 pseudomolecules only |
|---|---|---|---|
| total length (Mbp) | 177.4 | 177.2 | 174.5 |
| expected size (Mbp) | 205 | 205 | 205 |
| number of contigs | 1087 | 224 | 8 |
| N50 (Mbp) | 1.5 | 22.4 | 22.4 |
| complete BUSCOs (out of 1,375) | 1359 | 1346 | 1356 |
| complete and single copy BUSCOs (out of 1,375) | 1271 | 1300 | 1306 |
| complete and duplicated BUSCOs (out of 1,375) | 88 | 46 | 50 |
| fragmented BUSCOs (out of 1,375) | 5 | 8 | 6 |
| missing BUSCOs (out of 1,375) | 11 | 21 | 13 |

## Glucosinolate and myrosinase genes in the *E. cheiranthoides* genome

Three aliphatic glucosinolates – glucoiberverin (3-methylthiopropyl glucosinolate), glucoiberin (3-methylsulfinylpropyl glucosinolate), and glucocheirolin (3-methylsulfonylpropyl glucosinolate) – have been reported as the main glucosinolates in *E. cheiranthoides* (*Cole, 1976*; *Huang et al., 1993*). We confirmed their dominance in *E. cheiranthoides* var. *Elbtalaue*, but also identified additional aliphatic and indole glucosinolates at lower concentrations. By making use of the known glucosinolate biosynthetic genes from Arabidopsis (*Hull et al., 2000*; *Mikkelsen et al., 2000*; *Bak and Feyereisen, 2001*; *Bak et al., 2001*; *Kliebenstein et al., 2001b*; *Kroymann et al., 2001*; *Reintanz et al., 2001*; *Chen et al., 2003*; *Kroymann et al., 2003*; *Naur et al., 2003*; *Grubb et al., 2004*; *Mikkelsen et al., 2004*; *Piotrowski et al., 2004*; *Textor et al., 2004*; *Nozawa et al., 2005*; *Klein et al., 2006*;

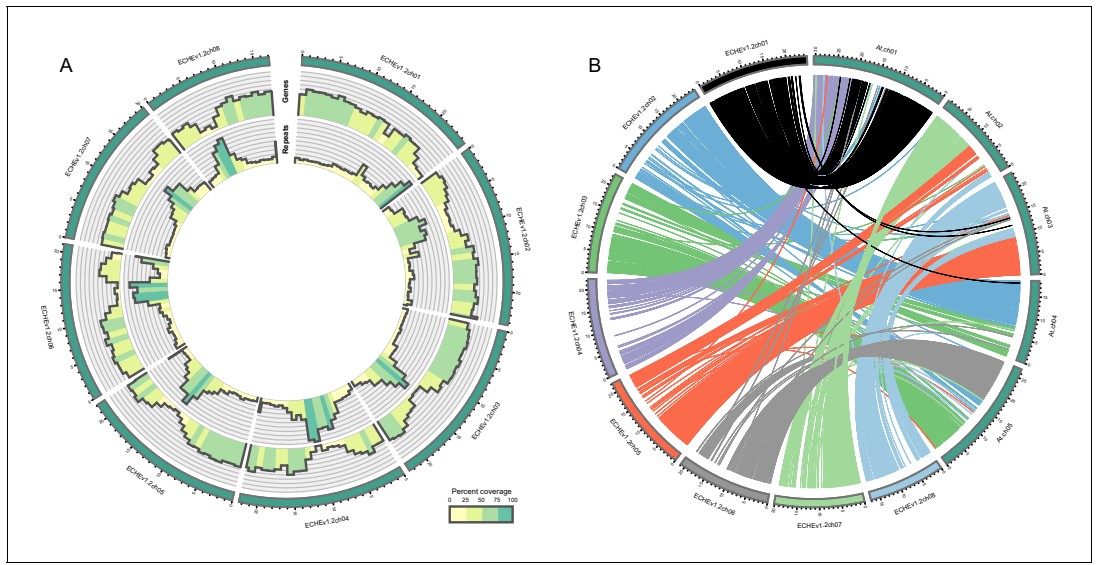

**Figure 2.** Visualization of the *E.cheiranthoides* genome assembly. (**A**) Circos plot of the *E. cheiranthoides* genome with gene densities (outer circle) and repeat densities (inner circle) shown as histogram tracks. Densities are calculated as percentages for 1 Mb windows. (**B**) Synteny plot of *E. cheiranthoides* and *A. thaliana*. Lines between chromosomes connect aligned sequences between the two genomes.

The online version of this article includes the following figure supplement(s) for figure 2:

**Figure supplement 1.** Post-clustering heatmap showing the density of Hi-C interactions between scaffolds used in the assembly.

**Figure supplement 2.** BUSCO completeness assessment for the EC1.2 genome assembly.

**Figure supplement 3.** BUSCO completeness assessment for the EC1.2 genome annotation.

**Figure supplement 4.** Proportional contributions of sequence classes to the genome sequence of *E. cheiranthoides*.

**Figure supplement 5.** Age distributions of intact LTR-RTs identified by LTR_retriever in the genome of *E. cheiranthoides*.

*Schuster et al., 2006*; *Hansen et al., 2007*; *Textor et al., 2007*; *Hansen et al., 2008*; *Knill et al., 2008*; *Li et al., 2008*; *Farquharson, 2009*; *Geu-Flores et al., 2009*; *Gigolashvili et al., 2009*; *He et al., 2009*; *Klein and Papenbrock, 2009*; *Knill et al., 2009*; *Pfalz et al., 2009*; *Sawada et al., 2009*; *He et al., 2010*; *Geu-Flores et al., 2011*; *He et al., 2011*; *Pfalz et al., 2011*; *Lächler et al., 2015*; *Kong et al., 2016*; *Pfalz et al., 2016*), BLASTn comparisons to the *E. cheiranthoides* genome, and creating phylogenetic trees to compare nucleotide coding sequences of Arabidopsis, *Brassica*, and *E. cheiranthoides*, we identified homologs of genes encoding both indole (*Figure 3*, *Figure 3— figure supplements 1–7*) and aliphatic (*Figure 4—figure supplements 1–8*) glucosinolate biosynthetic enzymes.

Homologs of all genes of the biosynthetic pathway for glucobrassicin (indol-3-ylmethyl glucosinolate) and its 4-hydroxy and 4-methoxy derivatives were present in *E. cheiranthoides* (*Figure 3*). Consistent with the absence of neoglucobrassicin (1-methoxy-indol-3-ylmethyl glucosinolate) in *E. cheiranthoides* var. *Elbtalaue*, we did not find homologs of the Arabidopsis genes encoding the biosynthesis of this compound.

Genes encoding the complete biosynthetic pathway of the *E. cheiranthoides* aliphatic glucosinolates glucoiberverin, glucoiberin, glucoerucin (4-methylthiobutyl glucosinolate), and glucoraphanin (4-methylsulfinylbutyl glucosinolate) were present in the genome (*Figure 4*, *Figure 4—figure supplements 1–6*). Because the *E. cheiranthoides* methylsulfonyl glucosinolates glucocheirolin, glucoerysolin (4-methylsulfonylbutyl glucosinolate), and 3-hydroxy-4-methylsulfonylbutyl glucosinolate are not present in Arabidopsis, genes encoding their biosynthesis are unknown and could not be identified as part of this study. The *E. cheiranthoides* genome contains genes with similarity to Arabidopsis *ALKENYL HYDROXALKYL PRODUCING* (*AOP2* and *AOP3*), and *3-BUTENYL GLUCOSINOLATE 2-HYDROXYLASE* (*GS-OH*) (*Figure 4—figure supplements 7* and *8*). However, the apparent absence of sinigrin (2-propenyl glucosinolate), 2-hydroxypropyl glucosinolate, progoitrin (2-hydroxy-3-butenyl glucosinolate), and 4-hydroxybutyl glucosinolate in *E. cheiranthoides* (*Figure 4*), suggests that the encoded enzymes of these genes have other functions. *CYP79A2*, which functions in the biosynthesis of benzyl glucosinolates that are present in very small amounts in seeds of Arabidopsis ecotype Columbia (*Wittstock and Halkier, 2000*), has a homolog in the *E. cheiranthoides* genome (*Figure 3—figure supplement 1*). Although we did not observe benzyl glucosinolates in *E. cheiranthoides*, this lack of detection could be due to assay sensitivity or not testing all tissue types. Homologs of the Arabidopsis *CYP79C1* and *CYP79C2* genes, which have unknown functions but are hypothesized to be involved in glucosinolate biosynthesis (*Halkier and Gershenzon, 2006*), are present in the *E. cheiranthoides* genome (*Figure 3—figure supplement 1*). *CYP79D2* from cassava catalyzed the formation of valine- and isoleucine-derived glucosinolates in Arabidopsis (*Mikkelsen and Halkier, 2003*), yet no *CYP79D* genes appear to be present in *E. cheiranthoides* (*Figure 3—figure supplement 1*). Additionally, there was no clear *E. cheiranthoides* homolog of *GLUCORAPHASATIN SYNTHASE 1* (*GRS1*, *Figure 4—figure supplement 9*), a 2-oxoglutarate-dependent dioxygenase that contributes to glucoraphasatin (4-methylthio-3-butenyl glucosinolate) biosynthesis in *R. sativus* (*Kakizaki et al., 2017*).

In response to insect feeding or pathogen infection, glucosinolates are activated by myrosinase enzymes (*Halkier and Gershenzon, 2006*). Between-gene phylogenetic comparisons revealed that homologs of known Arabidopsis myrosinases, the main foliar myrosinases *TGG1* and *TGG2* (*Xue et al., 1995*; *Barth and Jander, 2006*), root-expressed *TGG4* and *TGG5* (*Andersson et al., 2009*), and likely pseudogenes *TGG3* and *TGG6* (*Rask et al., 2000*; *Zhang et al., 2002*), were also present in the *E. cheiranthoides* genome (*Figure 4—figure supplement 10*). Additionally, we found homologs of the more distantly related Arabidopsis myrosinases *PEN2* (*Bednarek et al., 2009*; *Clay et al., 2009*) and *PYK10* (*Sherameti et al., 2008*; *Nakano et al., 2017*). In Arabidopsis, protein products of *epithiospecifier protein* (*ESP*), *epithiospecifier modifier* (*ESM*), and *nitrile specifier protein* (*NSP*) direct glucosinolate breakdown into nitriles, thiocyanates, or isothiocyanates (*Lambrix et al., 2001*; *Burow et al., 2006*; *Zhang et al., 2006*). Although we did not measure glucosinolate breakdown in *Erysimum*, we did find *ESP, ESM*, and *NSP* homologs in the *E. cheiranthoides* genome (*Figure 4—figure supplements 11* and *12*). Therefore, the pathway of glucosinolate activation appears to be largely conserved between Arabidopsis and *E. cheiranthoides*.

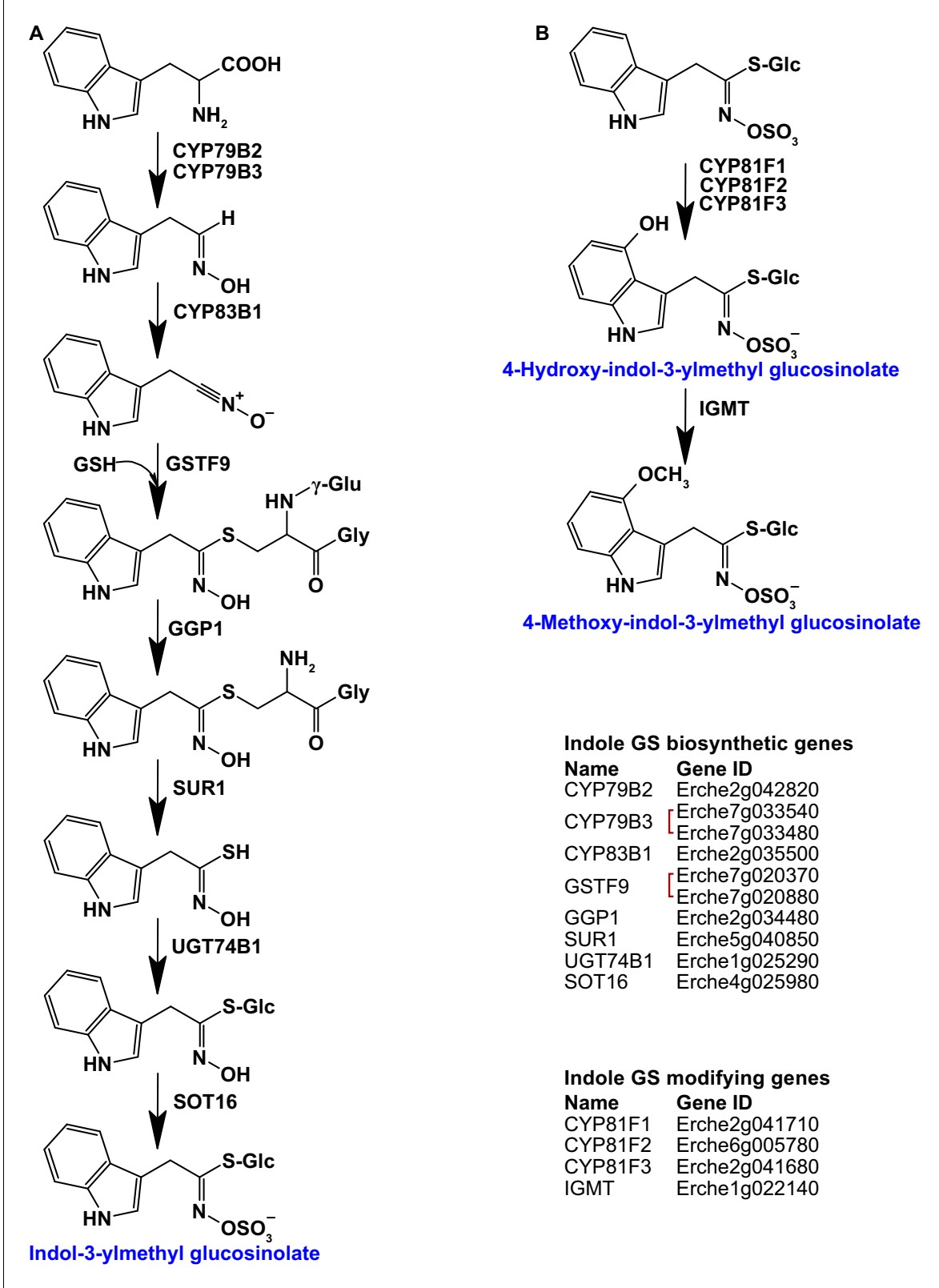

**Indole GS biosynthetic genes**

| Name | Gene ID |
|------|---------|
| CYP79B2 | Erche2g042820 |
| CYP79B3 | Erche7g033540 / Erche7g033480 |
| CYP83B1 | Erche2g035500 |
| GSTF9 | Erche7g020370 / Erche7g020880 |
| GGP1 | Erche2g034480 |
| SUR1 | Erche5g040850 |
| UGT74B1 | Erche1g025290 |
| SOT16 | Erche4g025980 |

**Indole GS modifying genes**

| Name | Gene ID |
|------|---------|
| CYP81F1 | Erche2g041710 |
| CYP81F2 | Erche6g005780 |
| CYP81F3 | Erche2g041680 |
| IGMT | Erche1g022140 |

**Figure 3.** Identification of known indole glucosinolate biosynthetic genes and glucosinolate-modifying genes from Arabidopsis in *Erysimum cheiranthoides*. (A) Starting with tryptophan, indole glucosinolates are synthesized using some enzymes that also function in aliphatic glucosinolate biosynthesis (GGP1; SUR1; UGT74B1) while also using indole glucosinolate-specific enzymes. (B) Indole glucosinolates can be modified by hydroxylation and subsequent methylation. Red square brackets indicate where gene copy numbers differ between Arabidopsis and *E. cheiranthoides*.
*Figure 3 continued on next page*

*Figure 3 continued*

Glucosinolates with names highlighted in blue were identified in *Erysimum cheiranthoides* var. *Elbtalaue*. Abbreviations: cytochrome P450 monooxygenase (CYP); glutathione *S*-transferase F (GSTF); glutathione (GSH); γ-glutamyl peptidase 1 (GGP1); SUPERROOT 1 C-S lyase (SUR1); UDP-dependent glycosyltransferase (UGT); sulfotransferase (SOT); glucosinolate (GS); indole glucosinolate methyltransferase (IGMT).

The online version of this article includes the following figure supplement(s) for figure 3:

**Figure supplement 1.** Phylogeny of cytochrome P450 monooxygenase (CYP) genes from *E. cheiranthoides*, *A. thaliana*, and *B. oleracea*.
**Figure supplement 2.** Phylogeny of glutathione *S*-transferase F (GSTF) and glutathione *S*-transferase Tau (GSTU) genes from *E. cheiranthoides*, *A. thaliana*, and *B. oleracea*.
**Figure supplement 3.** Phylogeny of γ-glutamyl peptidase 1 (GGP1) genes from *E. cheiranthoides*, *A. thaliana*, and *B. oleracea*.
**Figure supplement 4.** Phylogeny of SUPERROOT 1 C-S lyase (SUR1) genes from *E. cheiranthoides*, *A. thaliana*, and *B. oleracea*.
**Figure supplement 5.** Phylogeny of UDP-dependent glycosyltransferase (UGT) genes from *E. cheiranthoides*, *A. thaliana*, and *B. oleracea*.
**Figure supplement 6.** Phylogeny of sulfotransferase (SOT) genes from *E. cheiranthoides*, *A. thaliana*, and *B. oleracea*.
**Figure supplement 7.** Phylogeny of indole glucosinolate methyltransferase (IGMT) genes from *E. cheiranthoides*, *A. thaliana*, and *B. oleracea*.

## Phylogenetic relationship of 48 *Erysimum* species

Assemblies of transcriptomes from 48 *Erysimum* species (including *E. cheiranthoides*) had N50 values ranging from 574 to 2,160 bp (*Supplementary file 3*). Transcriptome assemblies contained completed genes from 54–94% of the BUSCO set and coding sequence lengths were generally shorter on average than the *E. cheiranthoides* coding sequence lengths (*Supplementary file 3*). Transcriptome sequences were deposited under GenBank project ID PRJNA563696 and at www.erysimum.org. The large number of orthologous gene sequences identified among the *E. cheiranthoides* genome and the 48 transcriptomes resulted in an ASTRAL species tree with high posterior probabilities for most nodes (*Figure 5—figure supplement 1*, *Supplementary file 4*). To determine divergence times among the 48 species, we generated a chronogram using a concatenated ExaML species tree with branch length information (*Figure 5*). While we relied on published estimates to constrain ages of several internal nodes, our analysis aligns well with a recent, rapid radiation of the species included in our study within the last 2–4 Mya (*Figure 5*).

The concatenated ExaML species tree and the ASTRAL species tree shared overall similar topologies, but very short internal branch lengths on both trees indicated high levels of gene tree discordance. We further dissected this discordance by assessing support of the main topology of the ASTRAL species tree (*Figure 5—figure supplement 1*) using quartet scores, which compare the main tree topology relative to its first and second alternative topology. For most nodes in the ingroup, each topology had quartet scores near the minimum value of 1/3 (*Supplementary file 4*), indicating that the possible gene trees were present in almost equal frequency for each topology. For the ExaML tree, we assessed discordance at each node using concordance factors, which are the proportion of gene trees that agree with the main topology (*Figure 5*). Again, most nodes in the ingroup had very low support for the main topology (<5% of gene tree agreement; *Figure 5*, *Supplementary file 5*). This suggests that many internal branches had lengths near 0, indicating polytomies that could not be resolved even with the extensive sampling of gene sequences from transcriptome data. These high levels of discordance were likely caused by frequent polyploidization (*Figure 5*), incomplete lineage sorting, and high degrees of hybridization. A high prevalence of hybridization was further indicated by a high frequency of gamma scores (hybridization proportion) between 0.3 and 0.7 across all ingroup taxa (*Figure 5—figure supplement 2*) recovered in the HyDe analysis (*Blischak et al., 2018*).

Despite extensive levels of discordance and low agreement of individual gene trees with the species trees, the main topologies of the ExaML and ASTRAL species trees revealed geographic clades that matched the generally limited native species ranges. The three Mediterranean annual species *E. incanum* (INC), *E. repandum* (REP), and *E. wilczekianum* (WIC) formed a well-supported monophyletic sister clade to all other sequenced species (*Figure 5*, *Figure 5—figure supplement 1*). The only other annual in the set of sampled species, *E. cheiranthoides* (ECE), was part of a weakly-supported clade (high posterior probability but low concordance), comprised of several perennial species from Greece and central Europe, including the widespread ornamental *E. cheiri* (CHR). Species from the Iberian peninsula/Morocco, North America, and Iran formed additional, weakly-supported clades conserved between species trees (*Figure 5*), while another clade of Turkish and Greek *Erysimum* species was only monophyletic in the ASTRAL species tree (*Figure 5—figure supplement 1*).

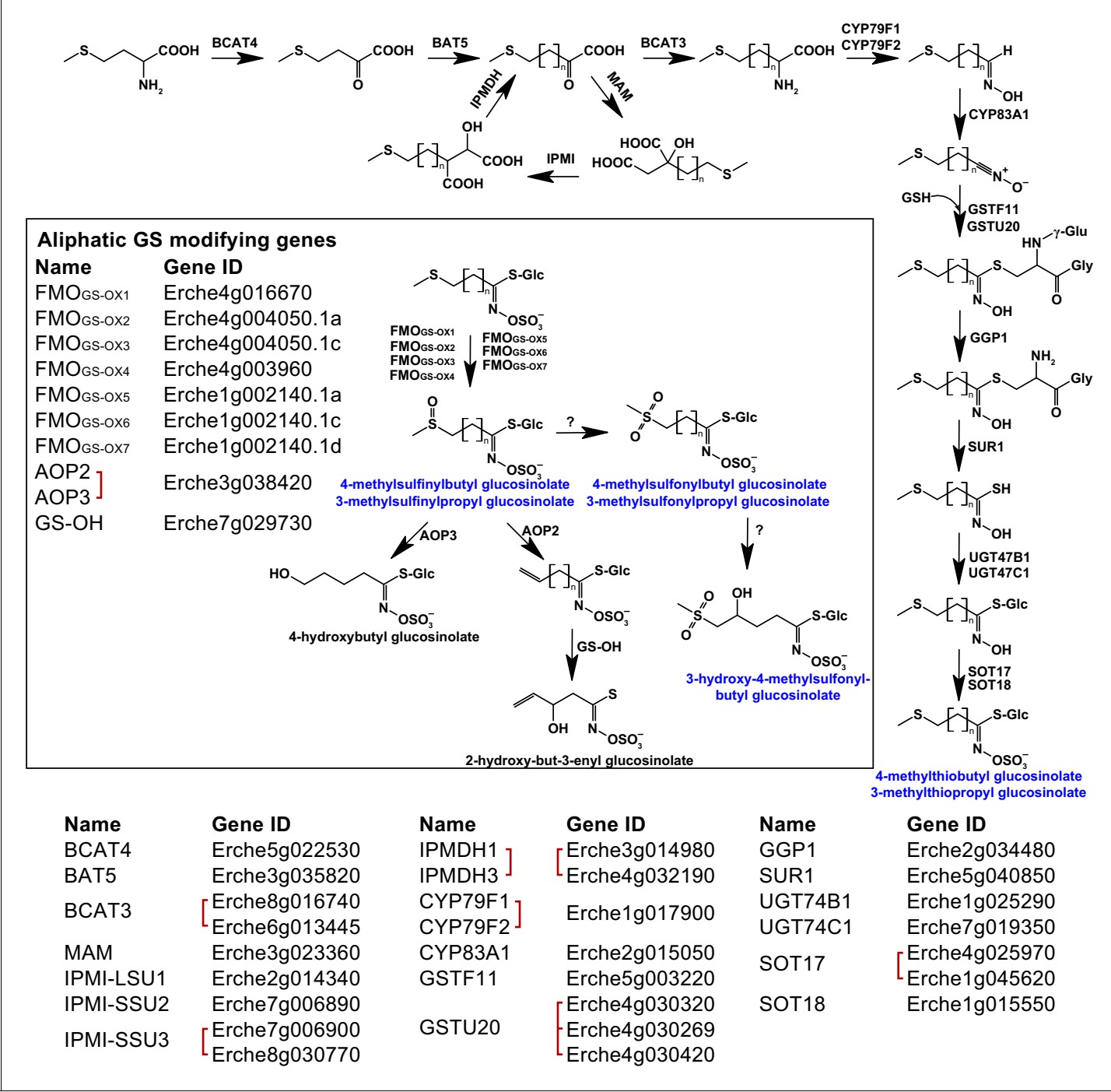

**Figure 4.** Identification of known aliphatic glucosinolate biosynthetic genes and glucosinolate-modifying genes from Arabidopsis in *Erysimum cheiranthoides*. Aliphatic glucosinolates are synthesized from methionine by a series of enzymes, while additional enzymes are responsible for aliphatic glucosinolate modifications (black box). Red square brackets indicate where gene copy numbers differ between Arabidopsis and *E. cheiranthoides*, or where gene copies could not be matched unambiguously between species. Glucosinolates with names highlighted in blue were identified in *Erysimum cheiranthoides* var. *Elbtalaue*. Abbreviations: branched-chain aminotransferase (BCAT); bile acid transporter (BAT); methylthioalkylmalate synthase (MAM); isopropylmalate isomerase (IPMI); large subunit (LSU); small subunit (SSU); isopropylmalate dehydrogenase(IPMDH); cytochrome P450 monooxygenase (CYP); glutathione *S*-transferase F (GSTF); glutathione *S*-transferase Tau (GSTU); glutathione (GSH); γ-glutamyl peptidase 1 (GGP1); SUPERROOT 1 C-S lyase (SUR1); UDP-dependent glycosyltransferase (UGT); sulfotransferase (SOT); flavin monooxygenase (FMO); glucosinolate oxoglutarate-dependent dioxygenase (AOP); 3-butenyl glucosinolate 2-hydroxylase (GS-OH).

The online version of this article includes the following figure supplement(s) for figure 4:

**Figure supplement 1.** Phylogeny of branched-chain aminotransferase (BCAT) genes from *E. cheiranthoides*, *A. thaliana*, and *B. oleracea*.

*Figure 4 continued on next page*

*Figure 4 continued*

**Figure supplement 2.** Phylogeny of bile acid transporter (BAT) genes from *E. cheiranthoides*, *A. thaliana*, and *B. oleracea*.

**Figure supplement 3.** Phylogeny of methylthioalkylmalate synthase (MAM) genes from *E. cheiranthoides*, *A. thaliana*, and *B. oleracea*.

**Figure supplement 4.** Phylogeny of isopropylmalate isomerase (IPMI) genes from *E. cheiranthoides*, *A. thaliana*, and *B. oleracea*.

**Figure supplement 5.** Phylogeny of isopropylmalate dehydrogenase (IPMDH) genes from *E. cheiranthoides*, *A. thaliana*, and *B. oleracea*.

**Figure supplement 6.** Phylogeny of flavin monooxygenase (FMO GS-OX) genes from *E. cheiranthoides*, *A. thaliana*, and *B. oleracea*.

**Figure supplement 7.** Phylogeny of glucosinolate oxoglutarate-dependent dioxygenase (AOP) genes from *E. cheiranthoides*, *A. thaliana*, and *B. oleracea*.

**Figure supplement 8.** Phylogeny of 3-butenyl glucosinolate 2-hydroxylase (GS-OH) genes from *E. cheiranthoides*, *A. thaliana*, and *B. oleracea*.

**Figure supplement 9.** Phylogeny of *E. cheiranthoides*, *A. thaliana*, and *B. oleracea* genes with similarity glucoraphasitin synthase (GRS) from *R. sativus*.

**Figure supplement 10.** Phylogeny of myrosinase (thioglucoside glucohydrolase, TGG) genes from *E. cheiranthoides*, *A. thaliana*, and *B. oleracea*.

**Figure supplement 11.** Phylogeny of epithiospecifier protein (ESP) and nitrile specifier protein (NSP) genes from *E. cheiranthoides*, *A. thaliana*, and *B. oleracea*.

**Figure supplement 12.** Phylogeny of epithiospecifier modifier (ESM) genes from *E. cheiranthoides*, *A. thaliana*, and *B. oleracea*.

The clear geographic structure in the main topologies of the species trees was confirmed by a strong correlation between the cophenetic and geographic distance matrices for the subset of 43 species with geographic information (Mantel test, p<0.001).

## Glucosinolate diversity and myrosinase activity

Across the 48 *Erysimum* species, we identified 25 candidate glucosinolate compounds with distinct molecular masses and HPLC retention times (*Supplementary file 6*). Of these, 24 compounds could be assigned to known glucosinolate structures with high certainty. The last remaining compound appeared to be an unknown isomer of glucocheirolin. Individual *Erysimum* species produced between 5 and 18 glucosinolates (*Figure 6A*), and total glucosinolate concentrations were highly variable among species (*Figure 6B*). The ploidy level of species explained a significant fraction of total variation in the number of glucosinolates produced ($F_{4,38} = 4.63$, p=0.004), with hexaploid species producing the highest number of compounds (*Figure 6—figure supplement 1*). However, neither the number of distinct glucosinolate compounds nor their total concentrations exhibited a phylogenetic signal, and related species were less similar than expected under a model of Brownian motion (*Table 2*).

Clustering species by dissimilarities in glucosinolate profiles mostly resulted in chemotype groups corresponding to known underlying biosynthetic genes, although support for individual species clusters in the chemogram was variable (*Figure 7*). The majority of all species produced glucoiberin as the primary glucosinolate. Of these, approximately half also produced sinigrin as a second dominant glucosinolate compound. Further chemotypic subdivision, related to the production of glucocheirolin and 2-hydroxypropyl glucosinolate, appeared to be present but only had relatively weak statistical support. However, eight species clearly differed from these general patterns. The species *E. allionii* (ALI), *E. rhaeticum* (RHA), and *E. scoparium* (SCO) mostly lacked glucosinolates with 3-carbon side-chains, but instead accumulated glucosinolates with 4-, 5- and 6-carbon side-chains. The two closely related species *E. odoratum* (ODO) and *E. witmannii* (WIT) predominantly accumulated indole glucosinolates, while *E. collinum* (COL), *E. pulchellum* (PUL), and accession ER2 predominantly produced glucoerypestrin (3-methoxycarbonylpropyl glucosinolate), a glucosinolate that is exclusively found within *Erysimum* (*Fahey et al., 2001*).

Similar to the lack of phylogenetic signal for compound numbers and concentrations, dissimilarity in glucosinolate profiles was unrelated to phylogenetic relatedness (Mantel test, p=0.331), and neither of the first two principal coordinates of the glucosinolate dissimilarity matrix showed a significant phylogenetic signal (*Table 2*). The lack of phylogenetic signal was visualized by optimizing vertical matching of tips between the ExaML species tree and the glucosinolate chemogram (*Figure 5—figure supplement 3A*). For five species pairs, the closest phylogenetic neighbor was also the most chemically similar species, but in general, close relatives more often belonged to chemically distant species clusters. Finally, reconstruction of the ancestral states for total glucosinolate content and the first principal coordinate of the glucosinolate dissimilarity matrix suggests that both traits likely originated at intermediate levels and repeatedly evolved towards opposite extremes in closely related species (*Figure 5—figure supplement 4*).

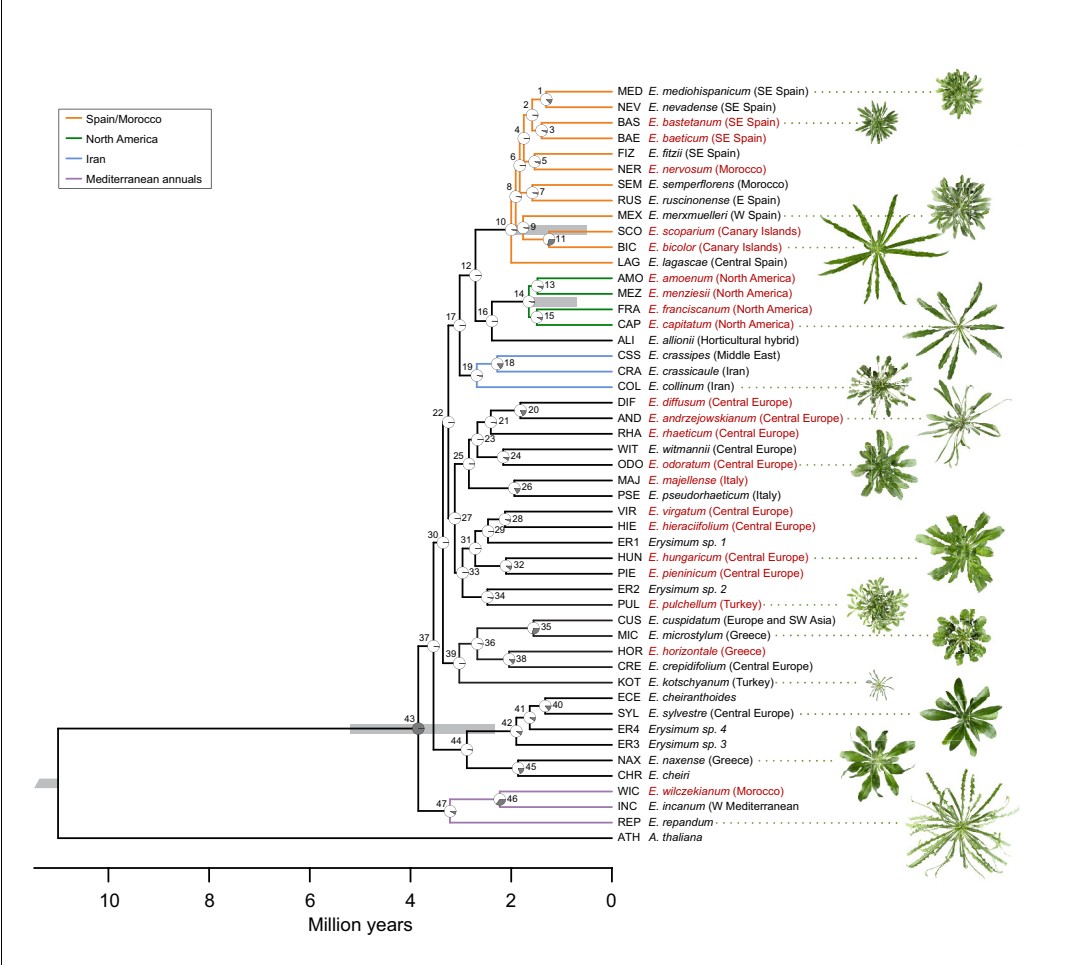

**Figure 5.** Genome-guided concatenated phylogeny of 48 *Erysimum* species. Phylogenetic relationships were inferred from 9868 orthologous genes using ExaML with *Arabidopsis thaliana* as outgroup. Node depth corresponds to divergence time in million years. Pie charts on each internode show concordance factors, with gray segments corresponding to the proportion of gene tree supporting the main topology. Nodes are labelled as 1 to 47; see **Supplementary file 5** for concordance factor values and number of decisive trees of each node. Four nodes were constrained using published divergence time estimates, with the range of constraints indicated by gray bars. Known polyploid species are highlighted in red. Approximate geographic range of species is provided in parentheses. The horticultural species *E. cheiri* and the weedy species *E. cheiranthoides* and *E. repandum* are of European origin but are now widespread across the Northern Hemisphere. Clades of species from shared geographic origins are highlighted in different colors. On the right, pictures of rosettes of a representative subset of species is provided to highlight the morphological diversity within this genus. Plants are of the same age and relative size differences are conserved in the pictures.

The online version of this article includes the following figure supplement(s) for figure 5:

**Figure supplement 1.** Genome-guided ASTRAL phylogeny of 48 *Erysimum* species.

**Figure supplement 2.** Hybridization levels among the 48 *Erysimum* species estimated by HyDe.

**Figure supplement 3.** Co-phylogenetic plots of optimized matches between phylogenetic relatedness and chemical similarity in (A) glucosinolates and (B) cardenolides.

**Figure supplement 4.** Ancestral state reconstruction of (A) total foliar glucosinolate content, (B) the first principal coordinate of the glucosinolate profile dissimilarity matrix, (C) total foliar cardenolide content, and (D) the first principal coordinate of the cardenolide profile dissimilarity matrix.

As glucosinolates require activation by myrosinase enzymes upon tissue damage by herbivores, myrosinase activity in leaf tissue determines the rate at which toxins are released. We quantified myrosinase activity of *Erysimum* leaf extracts and found it to be highly variable among species (*Figure 6C*). After grouping species into nine chemotypes defined by chemical dissimilarity and the production of characteristic glucosinolate compounds (*Figure 7C*), we found that myrosinase activity significantly differed among these chemotypes (*Figure 8*, $F_{8,33} = 8.31$, p<0.001). Chemotypes that predominantly accumulated methylsulfonyl glucosinolates, hydroxy glucosinolates, or indole

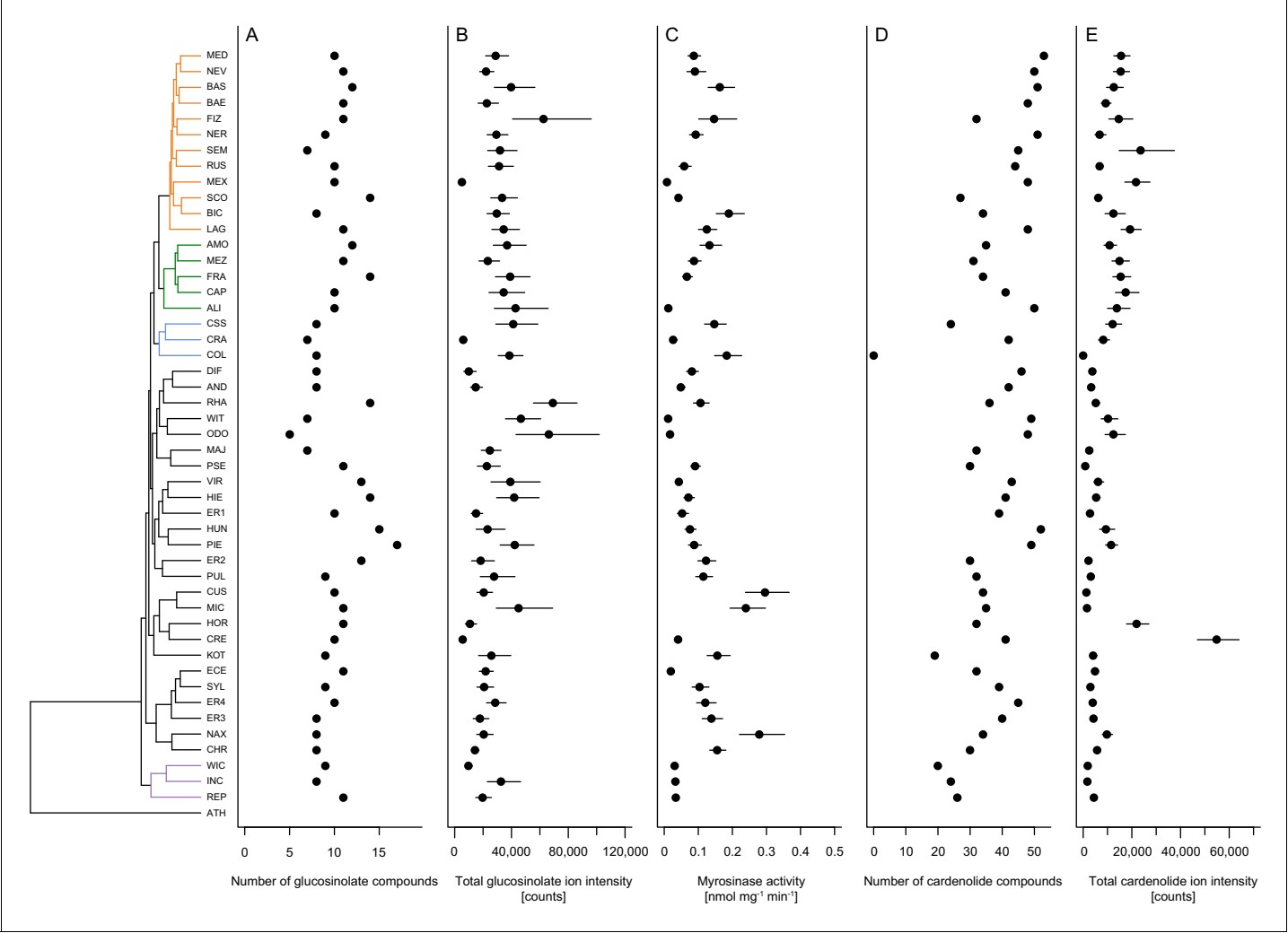

**Figure 6.** Mean defense traits of 48 *Erysimum* species, grouped by phylogenetic relatedness. Not all traits could be quantified for all species. (A) Total number of glucosinolate compounds detected in each species. (B) Total glucosinolate concentration found in each species, quantified by total ion intensity in the mass spectrometry analyses. Values are means ±1 SE. (C) Quantification of glucosinolate-activating myrosinase activity. Enzyme kinetics were quantified against the standard glucosinolate sinigrin (2-propenyl glucosinolate) and are expressed per unit fresh plant tissue. Values are means ±1 SE. (D) Total number of cardenolide compounds detected in each species. (E) Total cardenolide concentrations found in each species, quantified by total ion intensity in mass spectrometry analyses. Values are means ±1 SE.

The online version of this article includes the following source data and figure supplement(s) for figure 6:

**Source data 1.** Species means and standard errors (where applicable) for number of glucosinolate/cardenolide compounds, total compound concentrations, and myrosinase activity.

**Figure supplement 1.** Effect of ploidy on compound diversity.

**Figure supplement 2.** Correlation between cardenolide concentrations approximated by total cardenolide ion intensity and by inhibition of animal $Na^+/K^+$-ATPase.

glucosinolates had low to negligible activity against the assayed glucosinolate sinigrin. It is important to note that sinigrin is an alkenyl glucosinolate and *Erysimum* myrosinases targeting other, structurally dissimilar glucosinolates may not effectively cleave sinigrin. After chemotype differences were accounted for, myrosinase activity was marginally related to total glucosinolate concentrations ($F_{1,33}$ = 3.60, p=0.067). Similar to other glucosinolate traits, uncorrected myrosinase activity exhibited no phylogenetic signal (*Table 2*).

**Table 2.** Measure of phylogenetic signal for total defensive traits and principal coordinates of the cardenolide and glucosinolate dissimilarity matrices (PCO) using Blomberg's *K*. Significant values (p<0.05) are highlighted in bold.

| Plant trait | K statistics | p-value (10,000 simulations) |
|---|---|---|
| Glucosinolate PCO1 (18.8%) | 0.86 | **0.038** |
| Glucosinolate PCO2 (13.6%) | 0.80 | 0.090 |
| Total glucosinolate concentrations | 0.81 | 0.076 |
| Number of glucosinolate compounds | 0.89 | **0.014** |
| Myrosinase activity | 0.88 | **0.038** |
| Cardenolide PCO1 (16.6%) | 1.79 | **<0.001** |
| Cardenolide PCO2 (12.2%) | 1.04 | **0.002** |
| Total cardenolide concentrations | 1.03 | **0.015** |
| Number of cardenolide compounds | 1.25 | **<0.001** |

## Cardenolide diversity

With the exception of *E. collinum* (COL), which only contained trace amounts of cardenolides in leaves, all *Erysimum* species contained diverse mixtures of cardenolide compounds and accumulated considerable amounts of cardenolides (*Figure 6D–E*). The ploidy level of species again explained a significant fraction of the total variation in the number of cardenolides ($F_{4,38}$ = 3.47, p=0.016), with hexaploid species producing the highest average number of compounds (*Figure 6—figure supplement 1*). To obtain an estimate of biological activity and evaluate quantification from total MS ion counts, we used an established assay that quantifies cardenolide concentrations from specific inhibition of animal $Na^+/K^+$-ATPase by crude *Erysimum* leaf extracts. We found generally strong enzymatic inhibition, with leaves of *Erysimum* species on average containing an equivalent of 5.72 ± 0.12 µg $mg^{-1}$ (±1 SE) of the reference cardenolide ouabain. Despite only producing trace amounts of cardenolides, *E. collinum* (COL) extracts caused significantly stronger inhibition than the Brassicaceae control plant, *S. arvensis* (*Figure 6—figure supplement 2*). Overall, quantification of cardenolide concentrations by $Na^+/K^+$-ATPase inhibition was highly correlated with the total MS ion count (*Figure 6—figure supplement 2*, r = 0.95, p<0.001). Thus, the use of ion count data for cross-species comparisons was appropriate for this purpose. Both the total numbers of compounds and the total abundances exhibited a strong phylogenetic signal (*Table 2*), indicating that closely related species shared similar cardenolide traits.

Cardenolide diversity was considerably higher than that of glucosinolates, with a total of 97 distinguishable candidate cardenolide compounds identified across the 48 *Erysimum* species (two compounds were later excluded, leaving 95 compounds; *Supplementary file 7*). Of these, 46 compounds had distinct molecular masses and mass fragments, while the remaining compounds likely were isomers, sharing a molecular mass with other compounds but having distinct HPLC retention times. The 95 putative cardenolides comprised nine distinct genins (*Figure 9*, *Figure 9—figure supplement 1*), the majority of which were glycosylated with digitoxose, deoxy hexoses, xylose, or glucose moieties. Only digitoxigenin and cannogenol accumulated as free genins, while all other compounds occurred as either mono- or di-glycosides. A likely major source of isomeric cardenolide compounds was thus the incorporation of different deoxy hexoses of equivalent mass, such as rhamnose, fucose, or gulomethylose. A subset of compounds had molecular masses that were heavier by 42.011 m/z than known mono- or di-glycoside cardenolides. Such a gain in mass corresponds to the gain of an acetyl-group, and mass fragmentation patterns indicated that these compounds were acetylated on the first sugar moiety (*Supplementary file 7*). Out of the nine detected genins, six had previously been described from *Erysimum* species (*Makarevich et al., 1994*). In addition, we identified three previously undescribed mass features with fragmentation patterns characteristic of cardenolide genins (*Figure 9—figure supplement 1*). Of these three, one matched an acetylated cannogenol, one matched formylated cannogenol, and one matched formylated nigrescigenin, assuming acetylation/formylation of a free OH-group on the precursor molecule. Formyl adducts can sometimes be formed during LC-MS due to the addition of formic acid to solvents, although this is

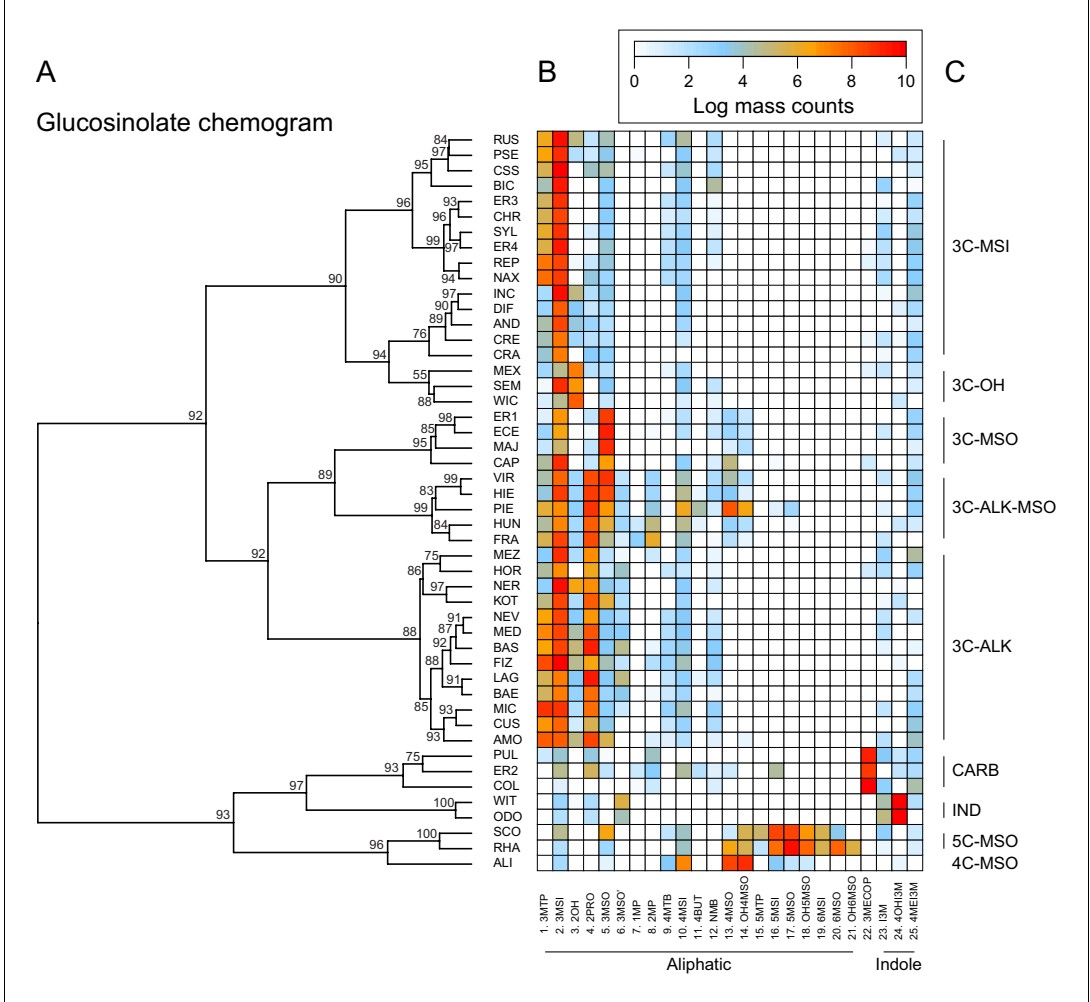

**Figure 7.** Glucosinolate compound diversity and abundance across 48 *Erysimum* species. (A) Chemogram clustering species by dissimilarities in glucosinolate profiles. Values at nodes are confidence estimates (approximately unbiased probability value, function *pvclust* in R) based on 10,000 iterations of multiscale bootstrap resampling. (B) Heatmap of glucosinolate profiles expressed by the 48 *Erysimum* species. Color intensity corresponds to log-transformed integrated ion counts recorded at the exact parental mass ([M-H]⁻) for each compound, averaged across samples from multiple independent experiments. Compounds are grouped by major biosynthetic classes and labelled using systematic short names. See ***Supplementary file 6*** for full glucosinolate names and additional compound information. (C) Classification of species chemotype based on predominant glucosinolate compounds. 3C/4C/5C = length of carbon side chain, MSI = methylsulfinyl glucosinolate, MSO = methylsulfonyl glucosinolate, OH = side chain with hydroxy group, ALK = side chain with alkenyl group, CARB = carboxylic glucosinolate, IND = indole glucosinolate.

less common with positive ionization. To exclude the possibility that these were technical artefacts, we analyzed a subset of extracts by LC-MS without the addition of formic acid and found both formyl-genins at comparable concentrations (***Figure 9—figure supplement 2***). We therefore assume that all three novel structures are natural variants of cardenolides produced by *Erysimum* plants, even though we currently lack final structural elucidation.

Clustering of species by dissimilarities in cardenolide profiles revealed fewer obvious species clusters in the chemogram than for glucosinolates, and particularly higher-level species clusters had only weak statistical support (***Figure 10***). A clear exception to this was a species cluster that included *E. cheiranthoides* (ECE) and *E. sylvestre* (SYL), which were characterized by a chemotype lacking several otherwise common cannogenol- and strophanthidin-glycosides, while accumulating unique digitoxi-genin-glycosides. A second major cluster was visually apparent, yet not statistically significant, and separated groups of species that did or did not produce glycosides of the newly discovered putative formyl-nigrescigenin (***Figure 10***). Similarity in cardenolide profiles among species was strongly corre-lated with phylogenetic relatedness (Mantel test, p<0.001), and the first two principal coordinates of

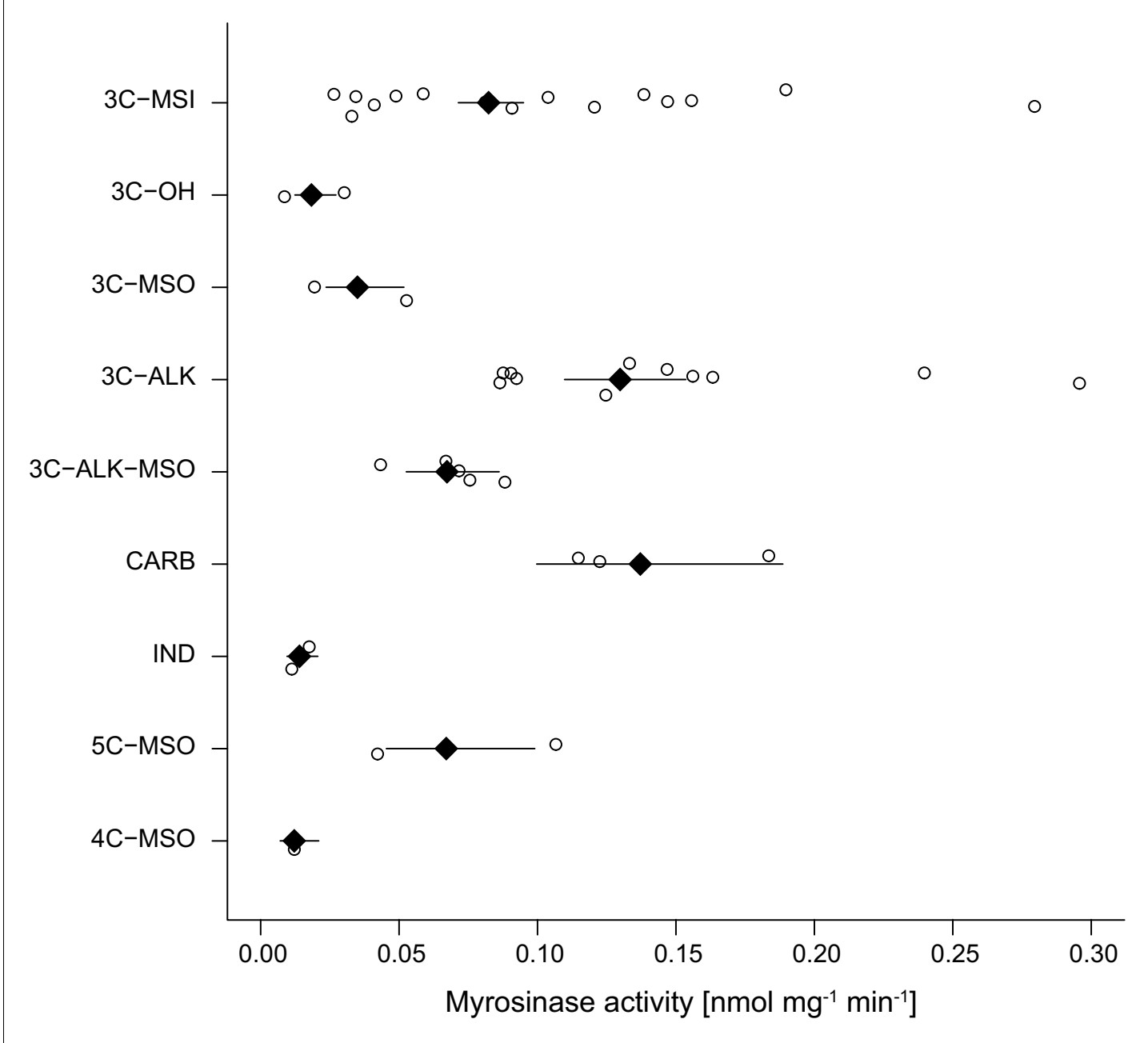

**Figure 8.** Myrosinase activity of leaf extracts from 43 *Erysimum* species, grouped by glucosinolate chemotype. Open circles are species means and black diamonds are chemotype means ± 1 SE. See also *Figure 7* for chemotype information. 3C/4C/5C = length of carbon side chain, MSI = methylsulfinyl glucosinolate, MSO = methylsulfonyl glucosinolate, OH = side chain with hydroxy group, ALK = side chain with alkenyl group, CARB = carboxylic glucosinolate, IND = indole glucosinolate.

The online version of this article includes the following source data for figure 8:

**Source data 1.** Species means for myrosinase activity and glucosinolate chemotype classification.

the Bray-Curtis dissimilarity matrix exhibited strong phylogenetic signals (*Table 2*). Closely related species were therefore not only more similar in their total cardenolide concentrations, but also had more similar cardenolide profiles than expected by chance. These results were again visualized by optimizing vertical matching of tips between the ExaML species tree and the cardenolide chemogram (*Figure 5—figure supplement 3B*). For twelve species pairs (half of all species in our phylogeny), the closest phylogenetic neighbor was also the most chemically similar species, and

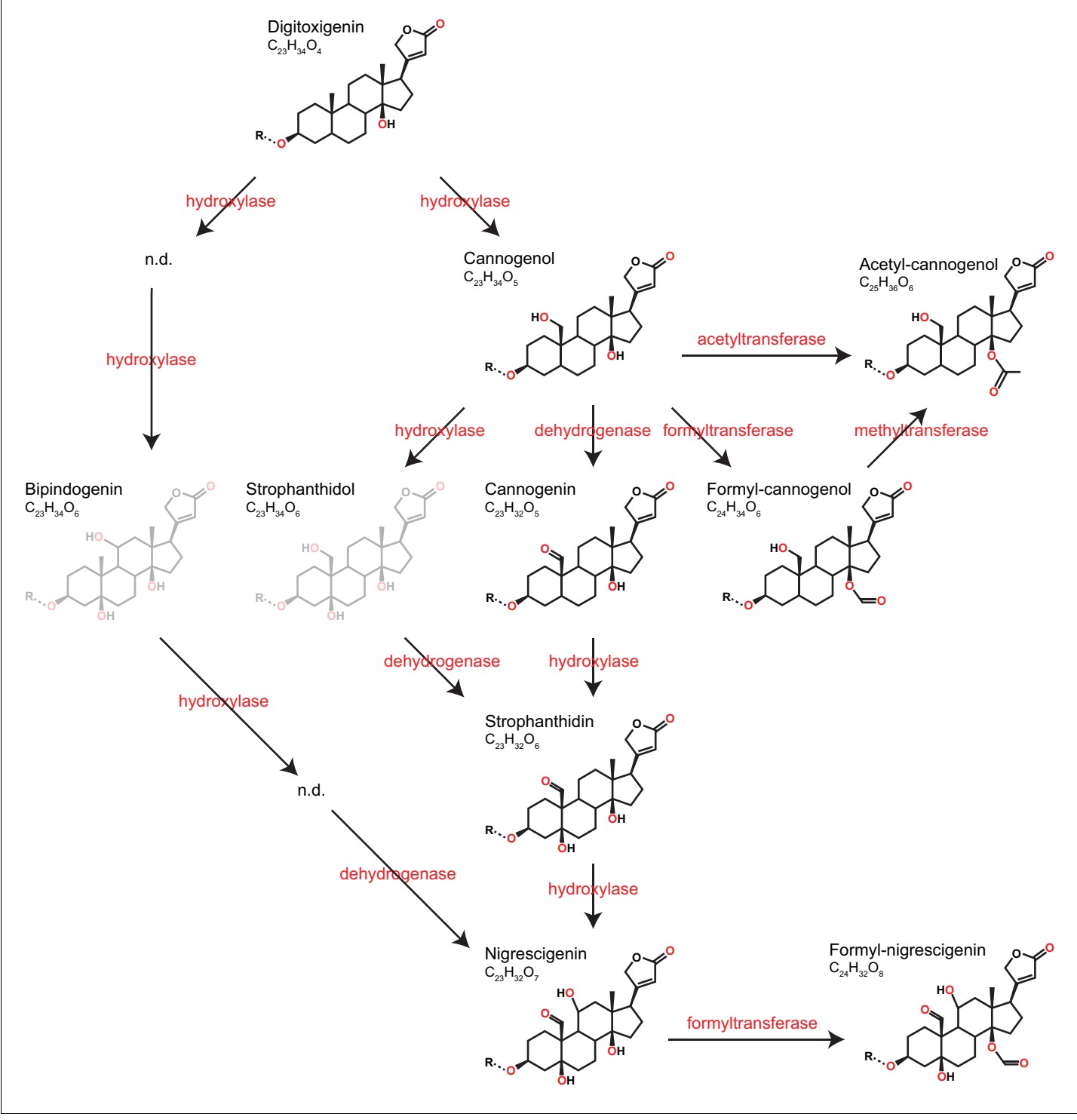

**Figure 9.** Predicted pathways of cardenolide genin modification in *Erysimum*. Pathways are linearized for simplicity but more likely form a complex network. Genin diversity likely originates from digitoxigenin, which is transformed into more stucturally complex cardenolides by hydroxylases, dehydrogenases, and formyl-, methyl-, or acetyltransferases. Acetyl-cannogenol could be derived directly from cannogenol or from formyl-cannogenol, with the frequent co-occurrence of acetyl-cannogenol and formyl-cannogenol in leaf extracts suggesting the latter. According to their exact mass, frequently detected dihydroxy-digitoxigenin compounds ($C_{23}H_{34}O_6$) could be either bipindogenin or strophanthidol (grayed out). While bipindogenin cardenolides have commonly been reported for *Erysimum* species in the literature, their structure would require additional intermediates that have not been detected (n.d.). Thus, strophanthidol appears to be the more likely isomer to occur in *Erysimum*. All cardenolide genins are further modified by

*Figure 9 continued on next page*

*Figure 9 continued*

glycosylation at a conserved position in the molecule (R). Note that all structures are putative, and particularly formyl- and acetyl-modifications could be attached to any free OH-group.

The online version of this article includes the following figure supplement(s) for figure 9:

**Figure supplement 1.** Characteristic MS fragmentation patterns of cardenolide genins used for putative identification of compounds.

**Figure supplement 2.** Effects of formic acid as a solvent additive in LC-MS analyses of cardenolide compounds.

phylogenetically related species more commonly belonged to chemically similar species clusters, indicated by a significantly lower total length of tip links compared to what was observed for glucosinolates (*Figure 5—figure supplement 3B*). Reconstruction of the ancestral states for total cardenolide content suggests that trait values likely originated at low total concentrations, and increased to intermediate levels in the North American and Spanish/Moroccan clades, and independently to very high levels in the species pair of *E. horizontale* and *E. crepidifolium* (*Figure 5—figure supplement 4*). For the first principal coordinate of cardenolide dissimilarity, trait values originated at intermediate values, but sub-clades more commonly evolved towards shared chemical profiles than was the case for glucosinolates (*Figure 5—figure supplement 4*).

## Macroevolutionary patterns in defense and inducibility

Similarity in glucosinolate and cardenolide chemical profiles of the 48 species was not correlated (Mantel test, p=0.171), and neither the number of compounds (PGLS: $F_{1,46} = 0.09$, p=0.771) nor their total concentrations (PGLS: $F_{1,46} = 0.51$, p=0.478) were correlated between compound classes. Tip-specific estimates of speciation rates were not correlated with the number of glucosinolate compounds produced by a species, regardless of speciation rate metric or statistical method used

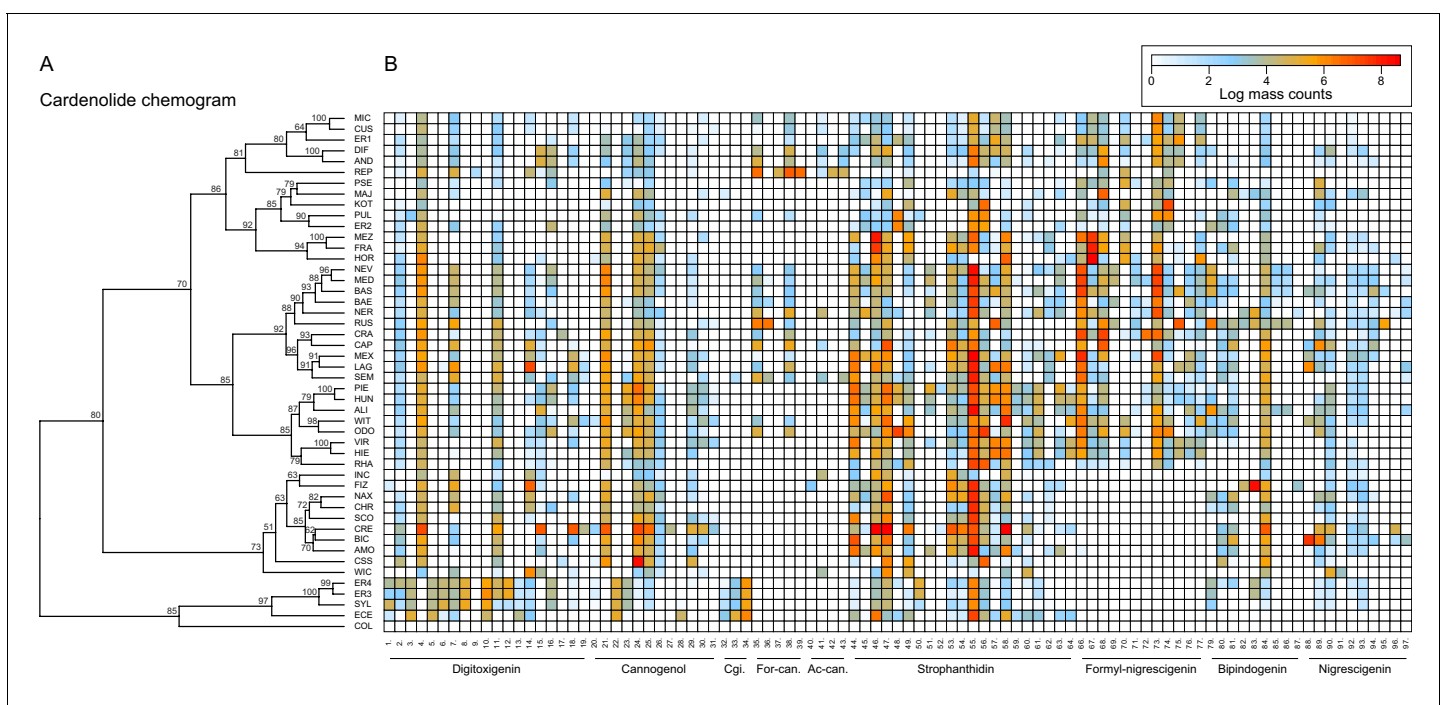

**Figure 10.** Cardenolide compound diversity and abundance across 48 *Erysimum* species. (**A**) Chemogram clustering species by dissimilarities in cardenolide profiles. Values at nodes are confidence estimates (approximately unbiased probability value, function *pvclust* in R) based on 10,000 iterations of multiscale bootstrap resampling. (**B**) Heatmap of cardenolide profiles expressed by the 48 *Erysimum* species. Color intensity corresponds to log-transformed integrated ion counts recorded at the exact parental mass ([M+H]$^+$ or [M+Na]$^+$, whichever was more abundant) for each compound, averaged across samples from multiple independent experiments. The species *E. collinum* (COL) only expressed trace amounts of cardenolides, which are not visible on the color scale. Compounds are grouped by shared genin structures. Cgi. = Cannogenin, For-can.=Formyl cannogenol, Ac-can. =Acetyl cannogenol. See *Supplementary file 7* for additional compound information.

(*Table 3*). In contrast, we found a significantly positive correlation between the node density (ND) measure and the number of cardenolide compounds, while for the alternate equal split (ES) measure the correlation was marginally significant for the simulation-based method only (*Table 3*). Given the correlation coefficients of the simulation-based method, variation in the number of cardenolide compounds thus explained 17–28% of the total variation in speciation rate. Variation in total glucosinolate or cardenolide concentrations was not correlated with speciation rates (*Table 3*).

Foliar application of JA was expected to stimulate accumulation of defensive compounds in plant leaves and among the 30 tested species, glucosinolate levels responded positively to JA, with the majority of species increasing their foliar glucosinolate concentration (*Figure 11*). However, the glucosinolate inducibility of a species was independent of constitutive glucosinolate levels (PGLS: $F_{1,28}$ = 0.17, p=0.680). By contrast, the majority of species exhibited lower cardenolide levels in response to JA, resulting in lack of inducibility across species (*Figure 11*). The species *E. crepidifolium* (CRE) heavily influenced inducibility patterns, as it not only had three times higher constitutive concentrations of cardenolides than any other *Erysimum* species, but also markedly increased both glucosinolate and cardenolide concentrations in response to JA treatment (*Figure 11*). When this outlier species was removed, inducibility (or suppression) of foliar cardenolides was not correlated with constitutive cardenolide levels (PGLS: $F_{1,27}$ = 0.20, p=0.657), and inducibilities of glucosinolates and cardenolides were likewise not correlated with each other (PGLS: $F_{1,27}$ = 0.36, p=0.551).

## Discussion

The genus *Erysimum* is a fascinating model system of phytochemical diversification that combines two potent classes of chemical defenses in the same plants. The assembled genome of the rapid-cycling annual plant *E. cheiranthoides* allowed us to identify almost the full set of genes involved in *E. cheiranthoides* glucosinolate biosynthesis, myrosinase expression, and breakdown product modification. This genome (GenBank project ID PRJNA563696, www.erysimum.org) will facilitate further identification of glucosinolate genes unique to *Erysimum* and represents a central resource for the identification of cardenolide biosynthesis genes in this emerging model system, as well as for future functional and evolutionary studies in the Brassicaceae.

The extant species diversity in the genus *Erysimum* is the result of a rapid radiation (*Moazzeni et al., 2014*), with our own estimate supporting an evolutionary recent onset of radiation within the last 2–4 Mya. In fact, as our approach for phylogeny construction did not account for heterozygosity levels of species, coalescence times are included in our divergence time estimates (*Edwards and Beerli, 2000*), thereby likely inflating age estimates for most speciation events. This may explain why we found no evidence for very recent speciation events (most recent event estimated at 1.24 Mya), while others have estimated significantly younger ages for at least some of the same events (*Moazzeni et al., 2014*, F. Perfectti, unpublished data).

All but one species in our study produced evolutionary novel cardenolides, while the likely closest relatives – the genera *Malcolmia*, *Physaria* or *Arabidopsis* (there is some disagreement among

**Table 3.** Correlations between plant traits and tip-specific speciation rates estimated from the main ExaML species tree.
Each trait is correlated against two estimates of speciation rates using phylogenetic least squares (PGLS) and a simulation-based method (SIM). Node density (ND) and equal split (ES) estimates of speciation rates are strongly correlated (r = 0.767, p<0.001) but differ in relative weighting of recent and more distant evolutionary history. For 1000 sets of randomly generated traits, only three resulted in more than one significant correlation, suggesting that multiple significant tests per trait (bold) are unlikely to arise by chance.

|                                     | Nd-pgls                        | Nd-sim                              | Es-pgls                        | Es-sim                              |
|-------------------------------------|--------------------------------|-------------------------------------|--------------------------------|-------------------------------------|
| Total glucosinolate concentrations  | $F_{1,46}$ = 0.79, p=0.379     | $r_{Pearson}$ = 0.29, p=0.300       | $F_{1,46}$ = 0.00, p=0.990     | $r_{Pearson}$ = 0.09, p=0.737       |
| Number of glucosinolate compounds   | $F_{1,46}$ = 1.16, p=0.286     | $r_{Pearson}$ = 0.23, p=0.412       | $F_{1,46}$ = 0.87, p=0.356     | $r_{Pearson}$ = 0.14, p=0.603       |
| Total cardenolide concentrations    | $F_{1,46}$ = 0.29, p=0.593     | $r_{Pearson}$ = 0.25, p=0.373       | $F_{1,46}$ = 0.01, p=0.908     | $r_{Pearson}$ = 0.23, p=0.398       |
| Number of cardenolide compounds     | **$F_{1,46}$ = 5.87, p=0.019** | **$r_{Pearson}$ = 0.53, p=0.030**   | $F_{1,46}$ = 1.93, p=0.17      | $r_{Pearson}$ = 0.42, p=0.093       |

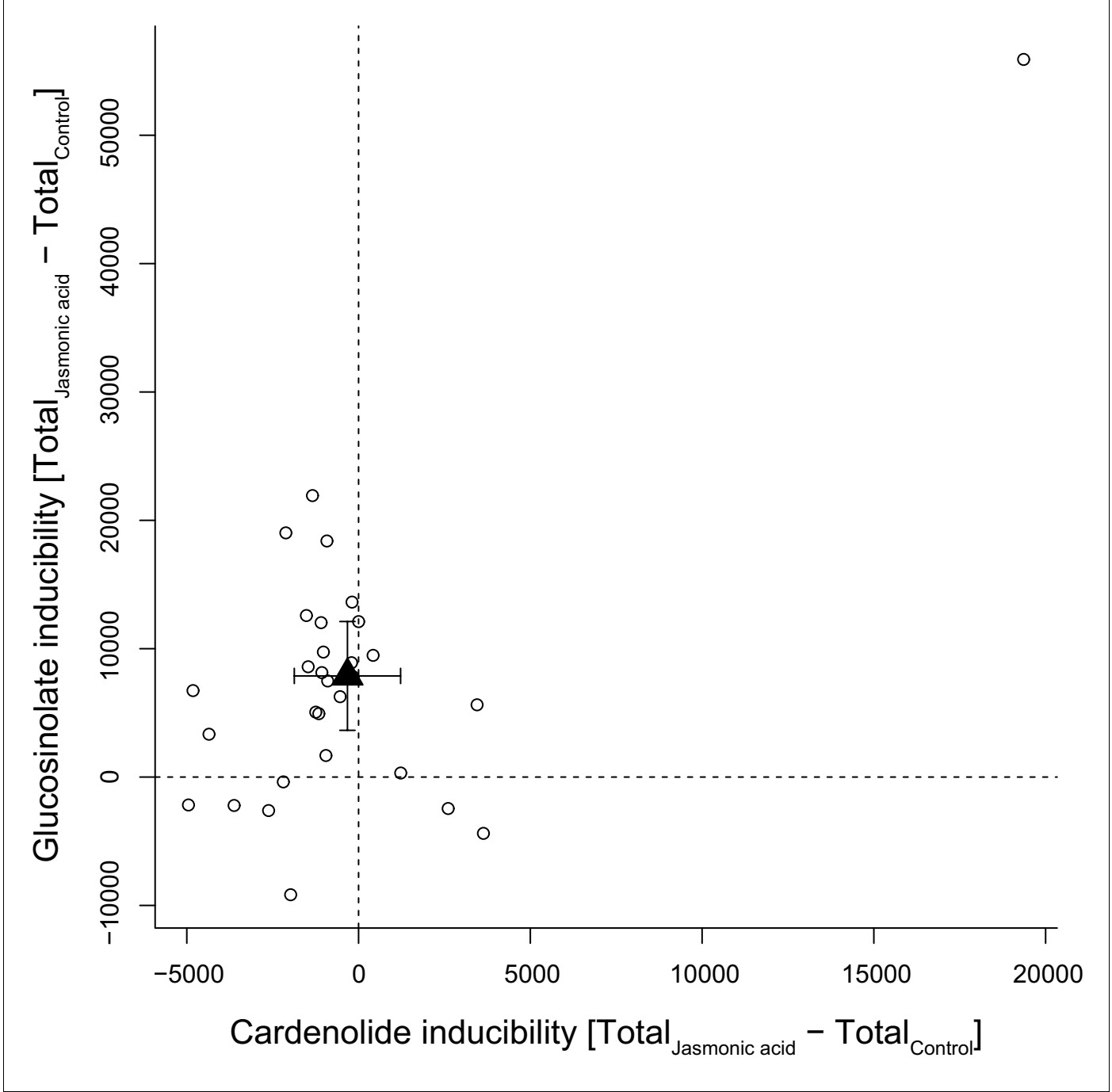

**Figure 11.** Inducibility of foliar glucosinolates and cardenolides in response to exogenous application of jasmonic acid (JA), expressed as absolute differences in total mass intensity between JA-treated and control plants. Circles are species means, based on single pooled samples of multiple individual plants. The filled triangle is the average inducibility of all measured species with 95% confidence interval. Non-overlap with zero (dashed lines) corresponds to a significant effect. The species in the upper right corner is *E. crepidifolium*, an outlier and strong inducer of both glucosinolates and cardenolides.

The online version of this article includes the following source data for figure 11:

**Source data 1.** Species means for absolute inducibility of total glucosinolate and cardenolide concentrations.

studies; *Moazzeni et al., 2014*; *Huang et al., 2016*) – almost certainly lack these defenses (*Jaretzky and Wilcke, 1932*; *Hegnauer, 1964*). The onset of diversification in *Erysimum* thus appears to coincide with the gain of the cardenolide defense trait, while the number of cardenolide compounds produced was positively correlated with tip-specific speciation rates of the main species tree, providing at least weak evidence that speciation and cardenolide diversification are linked in this genus. Even though most species co-expressed two different classes of potentially costly defenses, there was no evidence for a trade-off between glucosinolates and cardenolides, and both groups of traits varied independently.

Potentially costly, obsolete defenses are expected to be selected against and should disappear over evolutionary time. For example, cardenolides in the genus *Asclepias* and alkaloids across the Apocynaceae decrease in concentration with speciation, consistent with co-evolutionary de-escalation in response to specialized, sequestering herbivores (*Agrawal and Fishbein, 2008*; *Livshultz et al., 2018*). However, it appears that both glucosinolates and cardenolides provide a defensive function in *Erysimum*: glucosinolates may be maintained as highly efficient defenses against generalist herbivores (*Kerwin et al., 2015*), whereas cardenolides may be functionally relevant against glucosinolate-specialized herbivores (*Chew, 1975*; *Chew, 1977*; *Wiklund et al., 1978*; *Renwick et al., 1989*; *Dimock et al., 1991*).

As further evidence for the distinct roles of glucosinolates and cardenolides, the two defenses responded differently to exogenous JA application. Glucosinolate concentrations were upregulated in response to JA in the majority of species, with an average 52% increase relative to untreated controls. This is similar to the inducibility of glucosinolates reported for other Brassicaceae species (*Textor and Gershenzon, 2009*), suggesting that glucosinolate defense signaling remains unaffected by the presence of cardenolides in *Erysimum* plants. In contrast, cardenolide levels were not inducible or were even suppressed in response to exogenous application of JA in almost all tested species, suggesting that inducibility of cardenolides is not a general strategy of *Erysimum*. In the more commonly studied milkweeds (*Asclepias* spp., Apocynaceae), cardenolides are usually inducible in response to herbivore stimuli (*Rasmann et al., 2009*; *Bingham and Agrawal, 2010*), but cardenolide suppression is also common, particularly in plants with high constitutive cardenolide concentrations (*Bingham and Agrawal, 2010*; *Rasmann and Agrawal, 2011*). Milkweed plants are attacked by a rich community of cardenolide-specialized herbivores (*Dobler et al., 2012*), likely making this defensive plasticity adaptive for these plants. The lack of cardenolide inducibility in *Erysimum* could therefore indicate a lack of widespread cardenolide-specialized herbivores that might otherwise select against high constitutive cardenolide levels.

## Phylogenetic relationships and phytochemical similarity

The genus *Erysimum* poses considerable phylogenetic challenges, with its evolutionary recent radiation resulting in a large number of hybridizing species and high prevalence of polyploidization (*Polatschek and Snogerup, 2002*; *Marhold and Lihová, 2006*; *Al-Shehbaz, 2010*). Both previous partial phylogenies of the genus, constructed from internal transcribed spacer (ITS) or chloroplast sequences, consequently struggled to resolve polytomies among species (*Gómez et al., 2014*; *Moazzeni et al., 2014*).

Here, we attempted to construct a better-resolved phylogeny from 9868 orthologous genes extracted from transcriptome sequences. However, while our species tree provided good posterior probabilities for all nodes, it also revealed very high levels of gene discordance. Several internal nodes of our species tree topology were supported by less than 1% of all gene trees, and only the most recent branching events were supported by more than 10% of gene trees. High levels of discordance, likely driven by introgression and incomplete lineage sorting, are common during ongoing species radiations, with many recent plant examples reporting similar findings (*Novikova et al., 2016*; *Pease et al., 2016*; *Copetti et al., 2017*; *Wu et al., 2018*). The abundance of polyploid species in our phylogeny (at least 21 out of 48, *Figure 5*) may have further exacerbated levels of discordance. Specifically, if these are allopolyploid rather than autopolyploid species, discordance could be introduced by our methodological approach for gene selection, which randomly retained only a single copy for each identified orthologous gene with multiple copies. This same problem could also have inflated the estimation of hybridization by our HyDe analysis, although a high rate of gene flow is likely, at least among geographically close species (*Abdelaziz et al., 2014*). More fundamentally, the high levels of gene discordance also highlight the limitations of simple bifurcating species trees

to represent the significantly more complicated network of splits and reticulate evolutionary events that is likely the true history of the genus *Erysimum* (**Marhold and Lihová, 2006**). However, while we have been unsuccessful in reconstructing the exact evolutionary history of the genus, we nevertheless believe our species tree captures key aspects of its phylogenetic relationships, particularly in respect to closely related species and geographic clades.

In our species tree, the three annual species, *E. repandum*, *E. incanum*, and *E. wilczekianum* grouped together as a well-supported monophyletic clade sister to all other *Erysimum* species. These species co-occur geographically with several perennial *Erysimum* species, but they are largely isolated by non-overlapping flowering times. Further separate clades were present for species from Iran, North America, and a combined clade of species from Spain, Morocco, and the Canary Islands, while the remaining species grouped into four central and eastern European clades. Within the Spanish clade, species from southeastern Spain exhibited closer relatedness to Moroccan species than to species from northeastern or northwestern Spain, loosely matching more fine-scale evaluations of species relatedness in this region (**Abdelaziz et al., 2014**). Therefore, even though none of these clades were supported by high gene concordance factors, the main topology of our species tree captured an apparently meaningful pattern of closely related species commonly occurring in close geographic proximity.

Clustering of species by dissimilarities in glucosinolate profiles revealed distinct groups of chemically similar species, largely corresponding to nine chemotypes determined by the predicted function of few major-effect glucosinolate genes. However, we found no phylogenetic signal for chemical similarity, compound number, or total concentrations of glucosinolates (Blomberg's K < 1 for all traits). Closely related species thus appear to be less likely to share similar glucosinolate chemotypes. In a comparative study of 30 *Strepthanthus* species, **Cacho et al. (2015)** reported similarly low values for Blomberg's K for most glucosinolate traits, suggesting that this may be a general pattern in glucosinolate diversity.

In contrast, clustering of species by dissimilarities in cardenolide profiles revealed fewer distinct groups of chemically similar species, suggesting that the considerably more complex cardenolide chemotypes may be controlled by many minor-effect genes. However, we found strong phylogenetic signals for chemical similarity, compound numbers, and total concentrations of cardenolides (Blomberg's K > 1). Closely related species were therefore more likely to share similar cardenolide chemotypes. Given the high levels of gene discordance in our phylogeny, it seems probable that at least the glucosinolate results may have been affected by hemiplasy, i.e., a phenomenon where a trait is determined by genes whose topologies does not match the species tree (**Avise and Robinson, 2008**; **Pease et al., 2016**; **Wu et al., 2018**). Hemiplasy may result in the overestimation of a trait's evolutionary rate (**Mendes et al., 2018**), and results must be evaluated with caution. However, even though trait evolution within *Erysimum* almost certainly has been significantly more complex than can be represented by a simple bifurcating species tree (**Novikova et al., 2016**; **Pease et al., 2016**; **Wu et al., 2018**), we believe our results remain robust in three key points. First, geographically close *Erysimum* species tend to be more closely related, likely because geographic proximity may facilitate gene flow. Indeed, geographic signatures were present in both previously published phylogenies (**Gómez et al., 2014**; **Moazzeni et al., 2014**) as well as in our tree. Second, geographically close, related species share similar cardenolide but not glucosinolate phenotypes. Third, these distinct patterns in chemical classes indicate distinct evolutionary mechanisms for glucosinolate and cardenolide defenses, with a small number of major-effect glucosinolate genes evolving independently of a putatively larger set of minor-effect cardenolide genes.

## Phytochemical diversity

Despite vast morphological differences among sampled *Erysimum* species, the diversity in glucosinolate profiles across species was relatively limited, with a total of 25 different glucosinolate compounds detected. Even though this diversity may be further amplified by enzymes that direct glucosinolates into different toxic products upon activation (not measured here; **Lambrix et al., 2001**; **Burow et al., 2006**; **Zhang et al., 2006**), intraspecific glucosinolate diversity of Arabidopsis alone is significantly higher, with more than 30 compounds reported across a range of accessions (**Kliebenstein et al., 2001a**). In contrast, an evaluation of 30 *Streptanthus* species revealed a similarly low total number of 35 glucosinolate compounds (**Cacho et al., 2015**). The intraspecific diversity of Arabidopsis may therefore not be typical for all plants of the Brassicaceae. Additional broadly

comparative studies of glucosinolate diversity in other Brassicaceae species are needed to provide a reliable 'baseline' for glucosinolate diversity. Importantly, we intentionally ignored intraspecific chemical diversity as we lacked genetically diverse seed material for most species. However, a preliminary screening of multiple *E. cheiranthoides* accessions suggests that there is little to no variation in glucosinolate profiles within this one species, while there is considerable variation in cardenolide profiles (T. Züst, unpublished data).

The majority of *Erysimum* species produced glucoiberin as their main glucosinolate. Aliphatic glucosinolates such as glucoiberin are derived from methionine in a process that involves elongation and modification of a variable side-chain (*Halkier and Gershenzon, 2006*), and in this context the 3-carbon glucosinolate glucoiberin is one of the least biosynthetically complex glucosinolates. However, the potential to produce additional aliphatic glucosinolates with longer side chains clearly exists in the genus, as 4-, 5-, and 6-carbon glucosinolates with more complex modifications were scattered across the phylogeny. A few species produced glucosinolates that are not found in Arabidopsis, including a sub-class of aliphatic glucosinolates, the methylsulfonyl glucosinolates. The homolog of GS-OH, which in Arabidopsis forms 2-hydroxy-but-3-enyl glucosinolate from 3-butenyl glucosinolate, does not have a clear function in *E. cheiranthoides* due to the lack of alkenyl glucosinolates. It is therefore possible that the GS-OH homolog in *E. cheiranthoides* may code for the unknown enzyme that hydroxylates 4-methylsulfonylbutyl glucosinolate to form 3-hydroxy-4-methylsulfonylbutyl glucosinolate (*Figure 4*, *Figure 4—figure supplement 8*). Methylsulfonyl glucosinolates are found in several Brassicaceae genera (*Fahey et al., 2001*), and glucocheirolin, the most abundant methylsulfonyl glucosinolate in *Erysimum* species, is only a weak egg-laying stimulant for the cabbage white butterfly (*Pieris rapae*), compared to other glucosinolates (*Huang et al., 1993*). Methylsulfonyl glucosinolates may thus represent a plant response to specialist herbivores that use plant defenses as host-finding cues.

The species *E. pulchellum* (PUL) and *E. collinum* (COL) from Turkey and Iran, respectively, accumulated glucoerypestrin as their main glucosinolate compound. This compound was first described in *E. rupestre* [syn. *E. pulchellum*, (*Polatschek, 2011*)] by *Kjær et al. (1957)* and to date has been found exclusively in plants of the genus *Erysimum* (*Fahey et al., 2001*). Radioactive labeling experiments indicated that glucoerypestrin is derived from a dicarboxylic amino acid, possibly 2-amino-5-methoxycarbonyl-pentanoic acid (*Chisholm, 1973*). Modification of the amino acid side chain during methionine-derived aliphatic glucosinolate biosynthesis as a pathway to glucoerypestrin is less likely, due to the lower specific incorporation of $^{14}$C-labeled methionine compared to $^{14}$C-labeled dicarboxylic acids into this compound (*Chisholm, 1973*). In any case, the gain of glucoerypestrin represents yet another evolutionary novelty in the *Erysimum* genus, but its relative toxicity and the adaptive benefits of its production have yet to be elucidated.

Myrosinase activity levels differed among glucosinolate chemotypes, and activity was positively correlated with glucosinolate abundance in plants when controlling for glucosinolate chemotype. *Erysimum* species that predominantly produced indole glucosinolates or 4-methylsulfinyl glucosinolates had negligible myrosinase activity against the assayed aliphatic glucosinolate sinigrin. Indole glucosinolates can be activated by *PEN2* – a thioglucosidase that is more specific for indole glucosinolates (*Bednarek et al., 2009*; *Clay et al., 2009*) – or even break down in the absence of plant-derived myrosinase (*Kim et al., 2008*). The negligible activity in these species could therefore indicate the existence of selective pressures to tailor myrosinase expression to the type and concentrations of glucosinolates that are produced. Mirroring results for glucosinolate defenses, myrosinase activity was not more similar among related species, suggesting that the two components of glucosinolate defense evolve in concert.

We detected considerable amounts of the evolutionarily novel cardenolide defense in 47 out of 48 *Erysimum* species or accessions. Among the 95 likely cardenolide compounds detected, at least 22 had not been described previously in *Erysimum*. This metabolic diversity had three main sources: modification of the genin core structure, variation of the glycoside chain, or isomeric variation (e.g., through the incorporation of different isomeric sugars). Structural variation in cardenolides affects the relative inhibition of Na$^+$/K$^+$-ATPase (*Dzimiri et al., 1987*; *Petschenka et al., 2018*) and physiochemical properties such as lipophilicity, which play an important role in uptake and metabolism of plant metabolites by insects (*Duffey, 1980*). Individual *Erysimum* species produced between 15 and 50 different cardenolide compounds, and the comparison of quantification by total mass ion counts *vs.* quantification by inhibition of Na$^+$/K$^+$-ATPase revealed highly similar results. While both methods

of quantification are only approximate, this correlation at least provides no obvious indication of vast differences in $Na^+/K^+$-ATPase inhibitory activity among *Erysimum* cardenolides.

The metabolic pathways involved in the biosynthesis and modification of cardenolides have yet to be elucidated (*Kreis and Müller-Uri, 2010*; *Züst et al., 2018*). Here, we propose a pathway for the modification of digitoxigenin, commonly assumed to be the least biosynthetically complex cardenolide (*Kreis and Müller-Uri, 2010*), into the eight structurally more complex genins found within *Erysimum* (*Figure 9*). Variation in glycoside chains is likely mediated by glycosyltransferases that act on the different genins. In the Brassicaceae genus *Barbarea*, plants produce saponin glycosides as an evolutionary novel defense, and a significant proportion of glycoside diversity in this system has been linked to the action of a small set of UDP glycosyltransferases (*Erthmann et al., 2018*). Similarly, through the joint action of genin-modifying enzymes and glycosyltransferases, a relatively small set of enzymes and corresponding genes could generate the vast cardenolide diversity found in the *Erysimum* genus. The identification and manipulation of these genes in different *Erysimum* species will make it possible to test the adaptive benefits of this structural diversity.

On average, leaves of *Erysimum* species contained cardenolides equivalent to 6 µg ouabain per mg dry leaf weight (estimated from $Na^+/K^+$-ATPase inhibition), placing them slightly above most species of the well-studied cardenolide-producing genus *Asclepias* (*Rasmann and Agrawal, 2011*). However, two species, *E. collinum* (COL) and *E. crepidifolium* (CRE), were clear outliers in terms of cardenolide content (*Figure 6D–E*). The almost complete absence of cardenolides in *E. collinum* (COL), which clustered phylogenetically with two other Middle Eastern species producing average concentrations of these compounds (*E. crassipes* [CSS] and *E. crassicaule* [CRA], *Figure 5*), likely represents a secondary loss of this trait in the course of evolution. This species also accumulated an evolutionary novel glucosinolate, glucoerypestrin (see above), which may have resulted in a shift in selective pressures that led to the loss of potentially costly cardenolide production. Conversely, *E. crepidifolium* (CRE) had cardenolide concentrations more than three times higher than any other tested *Erysimum* species (*Figure 6E*). This is consistent with the highly toxic nature of this species, which has the German vernacular name 'Gänsesterbe' (geese death) and has been associated with mortality in geese that consume the plant.

Whereas most species did not induce cardenolide accumulation in response to JA, *E. crepidifolium* (CRE) had a significant 48% increase. While not as extreme, this observation is similar to the results of *Munkert et al. (2014)*, who reported a three-fold increase in cardenolide levels of *E. crepidifolium* in response to a high dose of methyl jasmonate. Plants use conserved transcriptional networks to continuously integrate signals from their environment and optimize allocation of resources to growth and defense (*Havko et al., 2016*). Thus, while these networks commonly govern hardwired responses (e.g., an attenuation of growth upon activation of JA signaling), they may nevertheless be altered by mutations at key nodes of the network (*Campos et al., 2016*). Given this relative flexibility in signaling networks, it is perhaps not surprising that the evolutionary novel cardenolides have been integrated into the defense signaling of *Erysimum* species to variable degrees. Investigating gene expression changes in the inducible *E. crepidifolium* relative to the non-inducing *E. cheiranthoides* may therefore provide valuable insights into the molecular regulation of this defense.

## Conclusions

The study of the speciose genus *Erysimum*, which has two co-expressed chemical defense classes, revealed largely independent evolution of the ancestral and the novel defense. With no evidence for trade-offs between the structurally and biosynthetically unrelated defenses, the diversity, abundance, and inducibility of each class of defenses appears to be evolving independently in response to the unique selective environment of each individual species. The evolutionarily recent gain of novel cardenolides has resulted in a system in which no known specific adaptations to cardenolides have yet evolved de novo in insect herbivores, although general adaptations to toxic food may still allow herbivores to consume the plants. *Erysimum* is thus an excellent model system for phytochemical diversification, as it facilitates the study of coevolutionary adaptations in real time. Our current work provides the foundation for a more mechanistic evaluation of these processes, which promises to greatly improve our understanding of the role of phytochemical diversity in plant-insect interactions.

# Materials and methods

## Plant material and growth conditions

The genus *Erysimum* is distributed across the northern hemisphere, with the center of diversity stretching from the Mediterranean Basin into Central Asia, and a smaller number of species centered in western North America (*Moazzeni et al., 2014*). Seeds of *Erysimum* species spanning a range of distributions in Europe and Western North America were collected in their native habitats or obtained from botanical gardens and commercial seed suppliers (*Figure 1*, *Supplementary file 1*). Ploidy levels of species were inferred from literature reports to test for the effect of ploidy on chemical diversity (*Supplementary file 1*). For seeds obtained from botanical gardens, we generally used species names as provided by the supplier. As an exception, seeds of *E. collinum* (COL) had originally been designated as *E. passgalense*, but these species names are now considered to be synonymous (*German, 2014*). Furthermore, plants of four seed batches did not exhibit the expected phenotypes and likely were the result of seed mislabeling by the suppliers. We nonetheless included these plants for transcriptome sequencing, but refer to them as accessions ER1, ER2, ER3, and ER4 (*Supplementary file 1*). For genome sequencing of *E. cheiranthoides*, seeds that were collected in 2015 by H. Christier from a natural population in the Elbe River floodplain (Germany, *Figure 1*) were planted in a greenhouse in one-liter pots in Cornell mix (by weight 56% peat moss, 35% vermiculite, 4% lime, 4% Osmocote slow-release fertilizer [Scotts, Marysville, OH], and 1% Unimix [Scotts]). This lineage, which we have designated '*Elbtalaue*', was propagated by self-pollination and single-seed descent for six generations prior to further experiments. Sixth-generation var. Elbtalaue seeds were submitted to the Arabidopsis Biological Resource Center (https://abrc.osu.edu) under accession number CS29250 and to the National Plant Germplasm System (https://www.ars-grin.gov/npgs/) under accession number PI 691911.

For transcriptome sequencing and metabolomic analyses of *Erysimum* species, subsets of the full species pool were grown in three separate experiments in 2016 and 2017. While some species were included in all three experiments, others could only be grown once due to limited seed availability or germination. To maximize germination success, seeds were placed on water agar (1%) in Petri dishes and cold-stratified for two weeks. After stratification, Petri dishes were moved to a growth chamber set to 24 ˚C day / 22˚C night at a 16:8 hr photoperiod. Viable seeds germinated within 3–10 days of placement in the growth chamber. As soon as cotyledons had fully extended, we transplanted the seedlings into 10 × 10 cm plastic pots filled with a mixture of peat-based germination soil (Seedlingsubstrat, Klasmann-Deilmann GmbH, Geeste, Germany), field soil, sand, and vermiculite at a ratio of 6:3:1:5. Plants were moved to a climate-controlled greenhouse set to 24 ˚C day / 16˚C night and 60% RH with natural light and supplemented artificial light set to a 14:10 hr photoperiod. Plants were watered as needed throughout the experiments, and fertilized with a single application of 0.1 L of fertilizer solution (N:P:K 8:8:6, 160 ppm N) three weeks after transplanting.

## *E. cheiranthoides* genome and transcriptome sequencing

DNA sequencing for genome assembly and RNA sequencing for annotation were conducted with samples prepared from sixth-generation inbred *E. cheiranthoides* var. *Elbtalaue*. High molecular weight genomic DNA was extracted from the leaves of a single *E. cheiranthoides* plant using Wizard Genomic DNA Purification Kit (Promega, Madison WI, USA). The quantity and quality of genomic DNA was assessed using a Qubit three fluorometer (Thermo Fisher, Waltham, MA, USA) and a Bioanalyzer DNA12000 kit (Agilent, Santa Clara, CA, USA). Twelve μg of non-sheared DNA were used to prepare the SMRTbell library, and the size-selection of 15–50 kb was performed on Sage BluePippin (Sage Science, Beverly, MA, USA) following manufacturer's instructions (Pacific Biosciences, Menlo Park, CA, USA) and as described previously (*Chen et al., 2019*). PacBio sequencing was performed by the Sequencing and Genomic Technologies Core of the Duke Center for Genomic and Computational Biology (Durham, NC, USA). For genome polishing, one DNA library was prepared using the PCR-free TruSeq DNA sample preparation kit following the manufacturer's instructions (Illumina, San Diego, CA), and sequenced on an Illumina MiSeq instrument (paired-end 2 × 250 bp) at the Cornell University Biotechnology Resource Center (Ithaca, NY).

The transcriptome of sixth-generation inbred *E. cheiranthoides* var. *Elbtalaue* plants was sequenced using both PacBio (Iso-Seq) and Illumina sequencing methods. Total RNA was isolated

from stems, flowers, buds, pods, young and mature leaves of five plants (siblings of the plant used for genome sequencing) using the SV Total RNA Isolation Kit with on-column DNAse I treatment (Promega, Madison, WI, USA). The RNA quantity and quality were assessed by RIN (RNA Integrity Number) using a 2100 Bioanalyzer (Agilent Technologies, Santa Clara, CA). The samples with a RIN value of >7 were pooled across all six tissue types. One µg of the pooled total RNA was used for the Iso-Seq following the manufacturer's instructions (Iso-Seq). The library preparation and sequencing were performed by Sequencing and Genomic Technologies Core of the Duke Center for Genomic and Computational Biology (Durham, NC, USA). For Illumina sequencing, 2 µg of purified pooled total RNA from three replicates was used for the preparation of strand-specific RNAseq libraries with 14 cycles of final amplification (*Zhong et al., 2011*). The purified libraries were multiplexed and sequenced with 101 bp paired-end read length in two-lanes on an Illumina HiSeq2500 instrument (Illumina, San Diego, CA) at the Cornell University Biotechnology Resource Center (Ithaca, NY). For Hi-C scaffolding, 500 mg of *E. cheiranthoides* leaf tissue was flash-frozen and sent to Phase Genomics (Phase Genomics Inc Seattle, WA, USA).

## *E. cheiranthoides* genome assembly and gene annotation

PacBio sequences from the genome of *E. cheiranthoides* were assembled using Falcon (*Chin et al., 2016*). The assembly was polished using Arrow from SMRT Analysis v2.3.0 (https://www.pacb.com/products-and-services/analytical-software/smrt-analysis/) with PacBio reads, and then assembled into chromosome-scale scaffolds using Hi-C methods by Phase Genomics (Seattle, WA, USA). Scaffolding gaps were filled with PBJelly v13.10 (*English et al., 2012*) using PacBio reads followed by three rounds of Pilon v1.23 correction (*Walker et al., 2014*) with 9 Gbp of Illumina paired-end 2 × 150 reads. BUSCO v3 (*Waterhouse et al., 2018*) metrics were used to assess the quality of the genome assemblies.

For gene model prediction, de novo repeats were predicted using RepeatModeler v1.0.11 (Smit, AFA, Hubley, R. *RepeatModeler Open-1.0*. 2008–2015 http://www.repeatmasker.org), known protein domains were removed from this set based on identity to UniProt (*Boutet et al., 2007*) with the ProtExcluder.pl script from the ProtExcluder v1.2 package (*Campbell et al., 2014*), and the output was then used with RepeatMasker v4-0-8 (Smit, AFA, Hubley, R and Green, P. *RepeatMasker Open-4.0*. 2013–2015 http://www.repeatmasker.org) in conjunction with the Repbase library. For gene prediction, RNA-seq reads were mapped to the genome with hisat2 v2.1.0 (*Kim et al., 2015*). Portcullis v1.1.2 (*Mapleson et al., 2018*) and Mikado v1.2.2 (*Venturini et al., 2018*) were used to filter the resulting bam files and make first-pass gene predictions. PacBio IsoSeq data were corrected using the Iso-Seq classify + cluster pipeline (*Gordon et al., 2015*). Augustus v3.2 (*Stanke et al., 2008*) and Snap v2.37.4ubuntu0.1 (*Korf, 2004*) were trained and then implemented through the Maker pipeline v2.31.10 (*Cantarel et al., 2008*) with Iso-Seq, proteins from Swiss-Prot, and processed RNA-seq added as evidence. Functional annotation was performed with BLAST v2.7.1+ (*Altschul et al., 1990*) and InterProScan v.5.36–75.0 (*Jones et al., 2014*).

## Repeat analysis

The genome of *E. cheiranthoides* was analyzed for LTR retrotransposons using LTRharvest (*Ellinghaus et al., 2008*), included in GenomeTools v1.5.10, with the parameters `-seqids yes -minlenltr 100 -maxlenltr 5000 -mindistltr 1000 -motif TGCA -motifmis 1 -maxdistltr 15000 -similar 85 -mintsd 4 -maxtsd 6 -vic 10 -seed 20 -overlaps best`. The genome was also analyzed using LTR_FINDER v1.07 (*Xu and Wang, 2007*) with parameters `-D 15000 -d 1000 -L 5000 -l 100 -p 20 -C -M 0.85 -w 0`. The results from LTRharvest and LTR_FINDER were then passed as inputs to LTR_retriever v2.0 (*Ou and Jiang, 2018*) using default parameters, including a neutral mutation rate set at $1.3 \times 10^{-8}$.

Using the LTR_retriever repeat library, the genome was masked with RepeatMasker v4.0.7, and additional repetitive elements were identified de novo in the genome using RepeatModeler. These repeats were used with blastx v2.7.1+ (*Altschul et al., 1990*) against the Uniprot and Dfam libraries and protein-coding sequences were excluded using the ProtExcluder.pl script from the ProtExcluder v1.2 package (*Campbell et al., 2014*). The masked genome was then re-masked with RepeatMasker, with the repeat library obtained from RepeatModeler. Coverage percentages for repeat types were

obtained using the fam_coverage.pl and fam_summary.pl scripts, which are included with LTR_retriever. All percentages were calculated based on the total length of the assembly.

## Genome-wide plot of genic sequence and repeats

A circular representation of the *E. cheiranthoides* genome was made with Circos v0.69–6 (*Krzywinski et al., 2009*). Gene and repeat densities were calculated by generating 1 Mb windows and by calculating percent coverage for the features using bedtools coverage v2.26.0 (*Quinlan and Hall, 2010*). The coverage values from the repeat library and the genome annotation were calculated independently of each other. Similarly, the total percentage of genic sequence for the genome was also calculated using bedtools genomecov v2.26.0. The gene and repeat percentages for the 1 Mb windows were then plotted as histogram tracks in Circos.

For analysis of synteny between *E. cheiranthoides* and *Arabidopsis thaliana* (Arabidopsis; TAIR10; www.arabidopsis.org), the genome sequences were aligned using NUCmer, from MUMmer v3.23 (*Kurtz et al., 2004*) with the parameters `–maxgap=500 –mincluster=100`. The alignments were filtered with delta-filter `-r -q -i 90 -l 1000` and coordinates of aligned segments were extracted with show-coords. The extracted coordinates were then used as the links input for Circos.

## Glucosinolate and myrosinase gene annotation in *E. cheiranthoides*

Glucosinolate and myrosinase genes in *E. cheiranthoides* were annotated based on homology to known Arabidopsis pathway genes in the Plant Metabolic Network database (https://pmn.plantcyc.org) (*Schläpfer et al., 2017*), as well as genes from *Brassica rapa* (*Wang et al., 2011*), *Raphanus sativus* (*Kakizaki et al., 2017*), and *Manihot esculenta* (*Mikkelsen and Halkier, 2003*). Coding sequences for the Arabidopsis and *Brassica* glucosinolate and myrosinase genes were obtained from GenBank (https://www.ncbi.nlm.nih.gov/genbank/), The Arabidopsis Information Resource (www.arabidopsis.org), and the *Brassica* Database (brassicadb.org/brad/glucoGene.php). Identified genes were used in a BLASTn query against the *E. cheiranthoides* coding sequence dataset. Coding sequences from *E. cheiranthoides*, Arabidopsis, and *Brassica* were used to construct maximum likelihood phylogenetic trees with 1000 bootstraps using MEGA version 7 (*Kumar et al., 2016*). Nucleotide sequences were aligned using MUSCLE, and the best model for each tree was selected based on neighbor-joining trees using the maximum likelihood method with gaps and missing data treated as a partial deletion with a 95% site coverage cut-off.

## Transcriptome sequencing of *Erysimum* species

To generate the large number of gene sequences required for a well-resolved phylogeny, we sequenced the foliar transcriptomes of 48 *Erysimum* species or accessions, including a first-generation inbred *E. cheiranthoides* var. *Elbtalaue*. Transcriptomes were generated from pooled leaf material of several individuals collected in the same experiment. Five species were sequenced from plants in experiment 2016, 18 species from experiment 2017–1, and 25 species from experiment 2017–2 (*Supplementary file 1*). Leaf material was harvested 5–7 weeks after plants were transplanted into soil. To average environmental and individual effects on RNA expression, we pooled leaf material from 2 to 5 individual plants from one or two time points (separated by 1–2 weeks) to create a single pooled RNA sample per species (see *Supplementary file 1* for details). For large-leaved species, we collected approximately 50 mg of fresh plant material from each harvested plant using a heat-sterilized hole punch (0.5 cm diameter). For smaller-leaved species, we collected an equivalent amount of material by harvesting multiple whole leaves. All leaf tissue was immediately snap frozen in liquid nitrogen and stored at −80°C until further processing. For sample pooling, we combined leaf material of individual plants belonging to the same species in a mortar under liquid nitrogen and ground all material to a fine powder. We then weighed out 50–100 mg of frozen pooled powder for each species.

We extracted RNA from pooled leaf material using the RNeasy Plant Mini Kit (Qiagen AG, Hombrechtikon, Switzerland), including a step for on-column DNase digestion, and following the manufacturer's instructions. The purified total RNA was dissolved in 50 µL RNase-free water, split into three aliquots, and stored at −80°C until further processing. Assessment of RNA quality, library preparation, and sequencing were all performed by the Next Generation Sequencing Platform of the University of Bern (Bern, Switzerland). RNA quality was assessed in one aliquot per extract using a

Fragment Analyzer (Model CE12, Agilent Technologies, Santa Clara, USA), and samples with low RIN scores (<7) were re-extracted and assessed again for quality. RNA libraries for TruSeq Stranded mRNA (Illumina, San Diego, USA) were assembled for each species and multiplexed in groups of eight, using unique index combinations (*Illumina, 2017*). Groups of eight multiplexed libraries were run together in single lanes (for a total of six lanes) of an Illumina HiSeq 3000 sequencer using 150 bp paired-end reads.

## De novo assembly of transcriptomes

RNA-seq data were cleaned with fastq-mcf v1.04.636 (https://github.com/ExpressionAnalysis/ea-utils/blob/wiki/FastqMcf.md) using the following parameters: `quality = 20, minimum read length = 50`. Filtered reads were assembled using Trinity v2.4.0 (*Haas et al., 2013*). The longest ORF was determined using TransDecoder v5.5.0 (https://github.com/TransDecoder). BUSCO v2 (*Waterhouse et al., 2018*) was run with lineage Embryophyta to assess gene representation and Orthofinder v2.3.1 (*Emms and Kelly, 2015*) was used to cluster proteins from all 48 transcriptomes into orthogroups.

## Phylogenetic tree construction

To construct a phylogenetic tree, we translated the assembled transcriptomes using TransDecoder v5.5.0. We then followed the Genome-Guided Phylo-Transcriptomics Pipeline (*Washburn et al., 2017*) to infer orthologous genes using synteny between genomes of *E. cheiranthoides* and Arabidopsis (TAIR10). Briefly, we obtained 26,830 orthologs between *E. cheiranthoides* and Arabidopsis through the syntenic blocks that were identified by SynMap from CoGe (https://genomevolution.org/coge/SynMap.pl). Sequences for each of the 48 *Erysimum* species and Arabidopsis (TAIR10_pep_20101214; www.arabidopsis.org) (total of 49 samples) were annotated using protein sequences of the orthologs using blastp v2.7.1 with an e-value $<10^{-4}$ and identity >85%. After annotation, single copy genes and one copy of repetitive genes were kept if they were present in more than 39 (>80% of 49) species. As a potential caveat, this approach would result in single copies from one parent of allopolyploid species to be retained randomly. In total, we recovered 11,890 genes, 9868 of which had orthologs with Arabidopsis. Each of these 9868 genes was aligned using MAFFT v7.394 (*Katoh et al., 2002*), and cleaned using Phyutility v2.2.6 (*Smith and Dunn, 2008*) with the parameter `-clean 0.3`. Maximum-likelihood tree estimation for each gene was constructed in RAxML v8.2.8 (*Stamatakis, 2014*) using the PROTCATWAG model with 100 bootstrap replicates. Finally, species tree inference was performed with these 9868 gene trees as input, using ASTRAL-III v5.6.3 (*Zhang et al., 2018*). In this species tree we observed very short internal branch lengths, which generally indicates high levels of gene tree discordance. We therefore additionally ran ASTRAL-III with the full annotation option (-*t 2*) for information on quartet support, total number of quartet trees in gene trees, and local posterior probabilities for the main topology and first and second alternative topologies. We additionally concatenated the cleaned alignments of the 9868 genes using the Genome-Guided Phylo-Transcriptomics Pipeline *concatenate_matrices.py* script (*Washburn et al., 2017*) to generate a completely random starting tree with RAxML v8.2.8, again using the PROTCATWAG model. Finally, we ran ExaML (*Kozlov et al., 2015*) to generate a concatenated phylogeny with parameters: `-a -B 100 m GAMMA -s`, and then evaluated the phylogeny with ExaML using `-f E -m GAMMA` and calculated gene concordance factors for each branch of the species tree using IQ-Tree v2 (*Minh et al., 2020*).

## Dating and hybridization detection

Using the ExaML concatenated phylogeny with branch length information, we dated the *Erysimum* radiation using treePL v1.0 (*Smith and O'Meara, 2012*). We constrained four nodes using inferred dates from two published studies. First, we constrained the node of the divergence between *Erysimum* species and Arabidopsis using a range of 11–22 Mya, determined by *Cardinal-McTeague et al. (2016)*. This range had been estimated using four fossil taxa and 151 extant taxa across the Brassicales, of which *Erysimum* is a member. Next, we constrained three additional nodes using dates determined by *Moazzeni et al. (2014)* based on a molecular clock approach with a published rate of substitution [$5 \times 10^{-9}$ substitutions per site per year, (*Kay et al., 2006*). We

constrained the node of the *Erysimum* clade to 2.33–5.2 Mya, the West European clade to 0.5–2.0 Mya, and the North American clade to 0.7–1.65 Mya.

To screen for evidence of hybridization we used HyDe v0.4.3 (*Blischak et al., 2018*) which tests sets of triples across the whole phylogeny. As HyDe requires the input file as a nucleotide alignment, we pulled the same set 9868 genes recovered during orthology determination from the original de novo assembly of transcriptomes. We then aligned and cleaned each gene sequence separately using MAFFT v7.394 (*Katoh et al., 2002*), and Phyutility v2.2.6 (*Smith and Dunn, 2008*) with the parameter `-clean 0.3`. A final concatenated alignment, used as input, of the 9868 genes was generated using the Genome-Guided Phylo-Transcriptomics Pipeline *concatenate_matrices.py* script (*Washburn et al., 2017*).

## Metabolite profiling of *Erysimum* leaves

We harvested leaf material for targeted metabolomic analysis of defense compounds from the same plants as used for transcriptome sequencing, one week after leaves for RNA extraction had been harvested. In each of the three experiments, we collected several leaves from 1 to 5 plants per species, and immediately snap froze the harvested leaves in liquid nitrogen. While most plant samples were screened for constitutive levels of chemical defenses only, we quantified inducibility of chemical defenses in a subset of 30 species with sufficient replication (eight or more plants) in the third experiment (2017–2). For these species, half of all plants were randomly assigned to the induction treatment and given a foliar spray of JA one week prior to harvest. Plants were sprayed with 2–3 mL of a 0.5 mM JA solution (Cayman Chemical, MI, USA) in 2% ethanol until all leaves were evenly covered in droplets on both sides. Control plants were sprayed with an equivalent amount of 2% ethanol solution. Harvested frozen plant material was lyophilized to dryness and ground to a fine powder. We weighed out 10 mg leaf powder per sample into separate tubes and added 1 mL of 70% MeOH extraction solvent. Samples were extracted by adding three 3 mm ceramic beads to each tube and shaking tubes on a Retsch MM400 ball mill three times for 3 min at 30 Hz. We centrifuged samples at 18,000 x g and transferred 0.9 mL of the supernatant to a new tube. Samples were centrifuged again, and 0.8 mL of the final supernatant was transferred to an HPLC vial for analysis by high-resolution mass spectrometry.

We analyzed extracts of individual plants (all experiments) or of multiple pooled individuals per species and induction treatment (experiment 2017–2 only) on an Acquity UHPLC system coupled to a Xevo G2-XS QTOF mass spectrometer with electrospray ionization (Waters, Milford MA, USA). Due to large differences in the physiochemical properties between glucosinolates and cardenolides, each plant extract was analyzed in two different modes to optimize detection of each compound class. For glucosinolates, extracts were separated on a Waters Acquity charged surface hybrid (CSH) C18 100 × 2.1 mm column with 1.7 µm pore size, fitted with a CSH guard column. The column was maintained at 40°C and injections of 1 µl were eluted at a constant flow rate of 0.4 mL/min with a gradient of 0.1% formic acid in water (A) and 0.1% formic acid in acetonitrile (B) as follows: 0–6 min from 2% to 45% B, 6–6.5 min from 45% to 100% B, followed by a 2 min wash phase at 100% B, and 2 min reconditioning at 2% B. For cardenolides, extracts were either separated using an equivalent method, or an optimized method using a Waters Cortecs C18 150 × 2.1 mm column with 2.7 µm pore size, fitted with a Cortecs C18 guard column. The column was maintained at 40°C and injections were eluted at a constant flow rate of 0.4 mL/min with a gradient of 0.1% formic acid in water (A) and 0.1% formic acid in acetonitrile (B) as follows: 0–10 min from 5% to 40% B, 10–15 min from 40% to 100% B, followed by a 2.5 min wash phase at 100% B, and 2.5 min reconditioning at 5% B.

Compounds were ionized in negative mode for glucosinolate analysis and in positive mode for cardenolide analysis. In both modes, ion data were acquired over an m/z range of 50 to 1200 Da in MS$^E$ mode using alternating scans of 0.15 s at low collision energy of 6 eV and 0.15 s at high collision energy ramped from 10 to 40 eV. For both positive and negative modes, the electrospray capillary voltage was set to 2 kV and the cone voltage was set to 20 V. The source temperature was maintained at 140°C and the desolvation gas temperature at 400°C. The desolvation gas flow was set to 1000 L/h, and argon was used as a collision gas. The mobile phase was diverted to waste during the wash and reconditioning phase at the end of each gradient. Accurate mass measurements were obtained by infusing a solution of leucine-enkephalin at 200 ng/mL at a flow rate of 10 µL/min through the LockSpray probe.

## Identification and quantification of defense compounds

Glucosinolates consist of a β-D-glucopyranose residue linked via a sulfur atom to a (Z)-N-hydroximi-nosulfate ester and a variable R group (*Halkier and Gershenzon, 2006*). We identified candidate glucosinolate compounds from negative scan data by the exact molecular mass of glucosinolates known to occur in *Erysimum* and related species (*Huang et al., 1993*; *Fahey et al., 2001*). In addition, we screened all negative scan data for characteristic glucosinolate mass fragments to identify additional candidate compounds (*Cataldi et al., 2010*). For mass features with multiple possible identifications, we inferred the most likely compound identity from relative HPLC retention times and the presence of biosynthetically related compounds in the same sample. We confirmed our identifications using commercial standards for glucoiberin, glucocheirolin (both Phytolab GmbH), and sinigrin (Sigma-Aldrich), as well as by comparison to extracts of *Arabidopsis* accessions with known glucosinolate profiles. Compound abundances of all glucosinolates were quantified by integrating ion intensities of the $[M-H]^-$ adducts using QuanLynx in the MassLynx software (v4.1, Waters).

All cardenolides share a highly conserved structure consisting of a steroid core (5β,14β-andros-tane-3β14-diol) linked to a five-membered lactone ring, which as a unit (the genin) mediates the specific binding of cardenolides to $Na^+/K^+$- ATPase (*Dzimiri et al., 1987*). While cardenolide genins are sufficient to inhibit $Na^+/K^+$- ATPase function, genins are commonly glycosylated or modified by hydroxylation on the steroid moiety to change the physiochemical properties and binding affinity of compounds (*Dzimiri et al., 1987*; *Petschenka et al., 2018*). We obtained commercial standards for the abundant *Erysimum* cardenolides erysimoside and helveticoside (Sigma-Aldrich), allowing us to identify these compounds through comparison of retention times and mass fragmentation patterns. Additional cardenolide compounds were tentatively identified from characteristic LC-MS fragmentation patterns. *Sachdev-Gupta et al. (1990)* and *Sachdev-Gupta et al. (1993)* reported fragmentation patterns for glycosides of strophanthidin, digitoxigenin, and cannogenol from *E. cheiranthoides*. Their results highlight the propensity of cardenolides to fragment at glycosidic bonds, with genin masses in particular being a prominent feature of cardenolide mass spectra. Additionally, cardenolide genins exhibit further fragmentation related to the loss of OH-groups from the steroidal core structure. We confirmed these rules of fragmentation for our mass spectrometry system using commercial standards of strophanthidin and digitoxigenin (Sigma-Aldrich). Importantly, while fragments were most abundant under high-energy conditions ($MS^E$), they were still apparent under standard MS conditions, likely due to in-source fragmentation.

Characteristic fragmentation allowed us to identify candidate cardenolide compounds in a genin-guided approach, where the presence of characteristic genin fragments in a chromatographic peak indicated the likely presence of a cardenolide molecule. We then identified the parental mass of these chromatographic peaks from the presence of paired mass features separated by 21.98 m/z, corresponding to the $[M+H]^+$ and $[M+Na]^+$ adducts of the intact molecule. For di-glycosidic carde-nolides, additional fragments corresponding to the loss of the outer sugar moiety allowed us to determine the mass and order of sugar moieties in the linear glycoside chain of the molecule. We screened our data for the presence of glycosides of strophanthidin, digitoxigenin, and cannogenol, and additional genins known to occur in *Erysimum* species (*Makarevich et al., 1994*). Multiple car-denolide genins can share the same molecular structure and may not be distinguished by mass spectrometry alone. Thus, all genin identifications are tentative and based on previous literature reports. We screened LC-MS data from two experiments (Phylo16 and Phylo17-2) to generate a list of carde-nolide compounds. Cardenolide data of experiment Phylo17-1 could not be used due to technical problems with the LC-MS analysis in positive mode. Relative compound abundances were quantified by integrating the ion intensities of the $[M+H]^+$ or the $[M+Na]^+$ adduct, whichever was more abundant for a given compound across all samples. In the third experiment, we added hydrocortisone (Sigma-Aldrich) to each sample as an internal standard, but between-sample variation (technical noise) was negligible compared to between-species variation.

For glucosinolate and cardenolide data separately, we averaged raw ion counts for each compound across experiments to yield a single chemical profile per compound class and species. We thereby intentionally ignored potential intraspecific variation in chemical diversity, as a lack of multiple seed origins or accessions for most species did not allow us to appropriately quantify this dimension of complexity. Raw ion counts were standardized by the dry sample weight, possible dilution of samples, and internal standard concentrations (where available). For pooled samples, ion counts

were standardized by the average dry weight calculated from all samples that contributed to a pool. The full set of standardized compound ion counts was then analyzed using linear mixed effects models (package *nlme* v3.1–137 in R v3.5.3). Because standardized ion counts still had a heavily skewed distribution, we applied a log(+0.1) transformation to all values. Log-transformed ion counts were modelled treating experiment as a fixed effect, and a species-by-compound identifier as the main random effect. Nested within the main random effect, we fitted a species-by-compound-by-experiment identifier as a second random effect to account for the difference of pooled or individual samples among experiments. The fixed effect of this model thus captures the overall differences in compound ion counts between experiments, while the main random effect captures the average deviation from an overall compound mean for each compound in each species. We extracted the overall compound mean and the main random effects from these models, providing us with average ion counts for each compound in each species on the log-scale. Negative values on the log-scale were set to zero as they would correspond to values below the limit of reliable detection of the LC-MS on the normal scale.

## Inhibition of mammal Na$^+$/K$^+$-ATPase by leaf extracts

Although all cardenolides target the same enzyme in animal cells, structural variation among different cardenolides can significantly influence binding affinity and thus affect toxicity (*Dzimiri et al., 1987*; *Petschenka et al., 2018*). Cardenolide quantification from LC-MS mass signal intensity does not capture such differences in biological activity, and furthermore may be challenging due to compound-specific response factors and narrow ranges of signal linearity. To evaluate whether total ion counts are an appropriate and biologically relevant measure for between-species comparisons of defense levels, we therefore quantified cardenolide concentrations by a separate method (*Züst et al., 2019*). For the subset of plants in the 2016 experiment, we measured the biological activity of leaf extracts on the Na$^+$/K$^+$-ATPase from the cerebral cortex of pigs (*Sus scrofa*, Sigma-Aldrich, MO, USA) using an in vitro assay introduced by *Klauck and Luckner (1995)* and adapted by *Petschenka et al. (2013)*. This colorimetric assay measures Na$^+$/K$^+$-ATPase activity from phosphate released during ATP consumption, and can be used to quantify relative enzymatic inhibition by cardenolide-containing plant extracts. Briefly, we tested the inhibitory effect of each plant extract at four concentrations to estimate the sigmoid enzyme inhibition function from which we could determine the cardenolide content of the extract relative to a standard curve for ouabain (Sigma Aldrich, MO, USA). We dried a 100 µL aliquot of each extract used for metabolomic analyses at 45°C on a vacuum concentrator (SpeedVac, Labconco, MO, USA). Dried residues were dissolved in 200 µL 10% DMSO in water, and further diluted 1:5, 1:50, and 1:500 using 10% DMSO. To quantify potential non-specific enzymatic inhibition that could occur at high concentrations of plant extracts, we also included control extracts from *Sinapis arvensis* leaves (a non-cardenolide producing species of the Brassicaceae) in these assays.

Assays were carried out in 96-well microplate format. Reactions were started by adding 80 µL of a reaction mix containing 0.0015 units of porcine Na$^+$/K$^+$-ATPase to 20 µL of leaf extracts in 10% DMSO, to achieve final well concentrations (in 100 µL) of 100 mM NaCl, 20 mM KCl, 4 mM MgCl$_2$, 50 mM imidazol, and 2.5 mM ATP at pH 7.4. To control for coloration of leaf extracts, we replicated each reaction on the same 96-well plate using a buffered background mix with identical composition as the reaction mix but lacking KCl, resulting in inactive Na$^+$/K$^+$-ATPases. Plates were incubated at 37°C for 20 min, after which enzymatic reactions were stopped by addition of 100 µL sodium dodecyl sulfate (SDS, 10% plus 0.05% Antifoam A) to each well. Inorganic phosphate released from enzymatically hydrolyzed ATP was quantified photometrically at 700 nm following the method described by *Taussky and Shorr (1953)*.

Absorbance values of reactions were corrected by their respective backgrounds, and sigmoid dose-response curves were fitted to corrected absorbances using a non-linear mixed effects model with a 4-parameter logistic function in the statistical software R (function *nlme* with *SSfpl* in package *nlme* v3.1–137).

$$Absorbance = \frac{A + (B - A)}{1 + e^{\left(\frac{(x_{mid} - x)}{scal}\right)}}$$

The absorbance values at four dilutions *x* are thus used to estimate the upper (*A*, fully active

enzyme) and lower ($B$, fully inhibited enzyme) asymptotes, the dilution value $x_{mid}$ at which 50% inhibition is achieved, and a shape parameter $scal$. In order to estimate four parameters from four absorbance values per extract, the $scal$ parameter was fixed for all extracts and changed iteratively to optimize overall model fit, judged by AIC. Individual plant extracts were treated as random effects to account for lack of independence within extract dilution series. For each extract we estimated $x_{mid}$ from the average model fit and the extract-specific random deviate. Using a calibration curve made with ouabain ranging from $10^{-3}$ to $10^{-8}$ M that was included on each 96-well plate, we then estimated the concentration of the undiluted sample in ouabain equivalents, i.e., the amount of ouabain required to achieve equivalent inhibition.

## Quantification of myrosinase activity

For the subset of plants in experiment 2017–2, we extracted the total amounts of soluble myrosinases from leaf tissue and quantified their activity as an important component of the glucosinolate defense system of these species. At the time of harvest of metabolomic samples, we collected an additional set of leaf disks from each plant, corresponding to approximately 50 mg fresh weight. After determination of exact fresh weight, samples were flash frozen in liquid nitrogen and stored at −80°C until enzyme activity measurements. Following the protocol of *Travers-Martin et al. (2008)*, frozen leaf material was ground and extracted in Tris-EDTA buffer (200 mM Tris, 10 mM EDTA, pH 5.5) and internal glucosinolates were removed by rinsing the extracts over a DEAE Sephadex A25 column (Sigma-Aldrich). Myrosinase activities were determined by adding sinigrin to plant extracts and monitoring the enzymatic release of glucose from its activation. Control reactions with sinigrin-free buffer were used to correct for plant-derived glucose. All samples were measured in duplicate and mean values related to a glucose calibration curve, measured also in duplicate. Reactions were carried out in 96-well plates and concentrations of released glucose were measured by adding a mix of glucose oxidase, peroxidase, 4-aminoantipyrine and phenol as color reagent to each well and measuring the kinetics for 45 min at room temperature in a microplate photometer (Multiskan EX, Thermo Electron, China) at 492 nm.

## Similarity in defense profiles between *Erysimum* species

To quantify chemical similarity among species, we performed separate cluster analyses on the glucosinolate and cardenolide profile data averaged across the three experiments. For each species, the log-transformed average ion counts of all compounds were converted to proportions (all compounds produced by a species summing to 1). From this proportional data we then calculated pairwise Bray-Curtis dissimilarities for all species pairs using function *vegdist* in the R package *vegan* v2.5–4 (*Oksanen et al., 2019*). We incorporated *vegdist* as a custom distance function for *pvclust* in the R package *pvclust* v2.0 (*Suzuki and Shimodaira, 2014*), which performs multiscale bootstrap resampling for cluster analyses. We constructed chemical dendrograms (chemograms) of glucosinolate and cardenolide profile similarities by fitting hierarchical clustering models (Ward's D) and estimated support for individual species clusters from 10,000 permutations. To test whether similarity in glucosinolate and cardenolide profiles was correlated with each other and with phylogenetic relatedness, we performed Mantel tests [function *mantel.test* in R package *ape* v5.0, (*Paradis and Schliep, 2019*) with 1000 permutations to compare Bray-Curtis dissimilarity matrices or the cophenetic distance matrix extracted for the ultrametric ExaML species tree. We visualized a potential correlation between the ExaML species tree and each chemogram by optimizing vertical matching of tree tips using function *cophylo* in the R package *phytools* v0.6–60 (*Revell, 2012*). In addition, we performed principal coordinate analyses (PCoA) on Bray-Curtis dissimilarity matrices of glucosinolate and cardenolide data using function *pcoa* in R package *ape* v5.0 and extracted the first two principal coordinates for each defense trait to test for phylogenetic signal.

## Phylogenetic signal in plant traits, ancestral state reconstruction, and trait correlations

We evaluated a prevalence of phylogenetic signal in chemical defense traits, myrosinase activity, and principal coordinates for both chemical dissimilarity matrices using Blomberg's $K$ (*Blomberg et al., 2003*). $K$ is close to zero for traits lacking phylogenetic signal; it approaches one if trait similarity among related species matches a Brownian motion model of evolution, and it can be >1 if similarity

is even higher than expected under a Brownian motion model. We estimated *K* for all traits using function *phylosig* in the *phytools* package and the ultrametric ExaML species tree. Additionally, we reconstructed the ancestral states for total glucosinolate and cardenolide content, and for the first principal coordinates of both chemical dissimilarity matrices using maximum-likelihood function *fastAnc* in the *phytools* package, which models the evolution of continuous traits using Brownian motion.

We tested for correlations among plant traits using phylogenetic generalized least squares (PGLS, function *gls* in the *nlme* package) with a Brownian correlation structure inferred from the ultrametric ExaML species tree. As variation in traits can impact the propensity for evolutionary diversification, we also tested for correlations between trait variation across the tips of the ExaML phylogeny and tip-specific estimates of speciation rates (tip-rate correlation, TRC) (*Harvey and Rabosky, 2018*). Tip-specific speciation rates were estimated as node density (ND) (*Freckleton et al., 2008*) and as inverse equal split (ES) measures (*Jetz et al., 2012*), where ND captures splitting dynamics over the entire history of a lineage leading to a tip, while ES is weighted towards branching patterns nearer the tips (*Harvey and Rabosky, 2018*). We tested for significant correlations between speciation rate measures and chemical traits using standard PGLS models, as well as a more robust simulation-based method (*Harvey and Rabosky, 2018*), thereby performing four tests for each of four chemical plant traits. To estimate the likelihood of false positives in this approach, we performed all four tests on 1000 sets of randomly generated plant traits and quantified the number of traits with more than one significant test. Finally, we evaluated the evidence for geographic signal in phylogenetic relatedness by performing a Mantel test to compare the cophenetic distance matrix to the pairwise geographic distances for all species with known collection locations.

## Acknowledgements

We thank Hartmut Christier for collecting *E. cheiranthoides* seeds, Erik Poelman for collecting *E. cheiri* seeds, and the botanical gardens listed in *Supplementary file 1* for providing seeds of additional species. Yvonne Künzi and Christoph Zwahlen assisted with the growth and maintenance of plants in the 48-species experiments, and with the harvesting and extraction of RNA and metabolomics samples. Sabrina Stiehler performed the $Na^+/K^+$-ATPase assay, and Jing Zhang helped with database maintenance. We thank Anurag Agrawal, Hamid Moazzeni, Gaurav Moghe, and Zephyr Züst for helpful advice and comments on the manuscript. Daniel Kliebenstein, Noah Whiteman, and Matthew W Hahn acted as reviewing editor and reviewers, respectively, and their comments helped to significantly improve our manuscript. This work was supported by Swiss National Science Foundation grant PZ00P3-161472 to TZ, a Triad Foundation grant to SRS, LAM, and GJ, US National Science Foundation awards 1811965 to CKH and 1645256 to GJ, Spanish AEI grant CGL2017-86626-C2-2-P and Junta de Andalucia Programa Operativo FEDER 2014–2020 A-RNM-505-UGR18 to FP, German Research Foundation grant DFG-PE 2059/3-1 to GP, and a grant within the LOEWE program (Insect Biotechnology and Bioresources) of the State of Hesse, Germany to GP.

## Additional information

### Funding

| Funder | Grant reference number | Author |
|---|---|---|
| Schweizerischer Nationalfonds zur Förderung der Wissenschaftlichen Forschung | PZ00P3-161472 | Tobias Züst |
| National Science Foundation | 1811965 | Cynthia K Holland |
| Triad Foundation | | Susan R Strickler<br>Lukas A Mueller<br>Georg Jander |
| National Science Foundation | 1645256 | Georg Jander |
| Deutsche Forschungsgemeinschaft | DFG-PE 2059/3-1 | Georg Petschenka |

| Agencia Estatal de Investiga-ción | CGL2017-86626-C2-2-P | Francisco Perfectti |
|---|---|---|
| LOEWE Program Insect Bio-technology and Bioresources | | Georg Petschenka |
| Junta de Andalucia Programa Operativo | FEDER 2014-2020 A-RNM-505-UGR18 | Francisco Perfectti |

The funders had no role in study design, data collection and interpretation, or the decision to submit the work for publication.

## Author contributions

Tobias Züst, Conceptualization, Data curation, Formal analysis, Supervision, Funding acquisition, Validation, Investigation, Visualization, Methodology, Writing - original draft, Project administration, Writing - review and editing; Susan R Strickler, Data curation, Software, Formal analysis, Visualization, Methodology, Writing - review and editing; Adrian F Powell, Software, Formal analysis, Visualization, Methodology, Writing - review and editing; Makenzie E Mabry, Hong An, Software, Formal analysis, Writing - review and editing; Mahdieh Mirzaei, Cynthia K Holland, Formal analysis, Investigation, Visualization, Writing - review and editing; Thomas York, Software, Formal analysis, Methodology, Writing - review and editing; Pavan Kumar, Formal analysis, Investigation, Writing - review and editing; Matthias Erb, Conceptualization, Resources, Methodology, Writing - review and editing; Georg Petschenka, Caroline Müller, Resources, Methodology, Writing - review and editing; José-María Gómez, Francisco Perfectti, Conceptualization, Resources, Writing - review and editing; J Chris Pires, Resources, Supervision, Methodology, Writing - review and editing; Lukas A Mueller, Resources, Data curation, Supervision, Writing - review and editing; Georg Jander, Conceptualization, Resources, Supervision, Funding acquisition, Methodology, Writing - original draft, Project administration, Writing - review and editing

## Author ORCIDs

Tobias Züst  https://orcid.org/0000-0001-7142-8731
Pavan Kumar  https://orcid.org/0000-0003-4718-3601
Matthias Erb  http://orcid.org/0000-0002-4446-9834
Georg Petschenka  http://orcid.org/0000-0002-9639-3042
Francisco Perfectti  http://orcid.org/0000-0002-5551-213X
Georg Jander  http://orcid.org/0000-0002-9675-934X

## Decision letter and Author response

Decision letter https://doi.org/10.7554/eLife.51712.sa1
Author response https://doi.org/10.7554/eLife.51712.sa2

# Additional files

## Supplementary files

- Supplementary file 1. Origin of *Erysimum* species and seed material.
- Supplementary file 2. Repetitive sequences and transposable elements in the *E. cheiranthoides* genome.
- Supplementary file 3. Transcriptome assembly metrics.
- Supplementary file 4. Discordance metrics for the ASTRAL phylogeny.
- Supplementary file 5. Discordance metrics for the ExaML phylogeny.
- Supplementary file 6. List of identified glucosinolate compounds.
- Supplementary file 7. List of identified cardenolide compounds.
- Transparent reporting form

## Data availability

Sequence data are available under GenBank project ID PRJNA563696 and www.erysimum.org, while all trait data and R code for trait and phylogenetic analyses are available from the Dryad Digital Repository.

The following datasets were generated:

| Author(s) | Year | Dataset title | Dataset URL | Database and Identifier |
|---|---|---|---|---|
| Züst T, Strickler SR, Powell AF, Mabry ME, An H, Mirzaei M, York T, Holland CK, Kumer P, Erb M, Petschenka G, Gómez J-M, Perfectti F, Müller C, Pires JC, Mueller LA, Jander G | 2019 | Data from: Rapid and independent evolution of ancestral and novel defenses in a genus of toxic plants (Erysimum, Brassicaceae) | http://dx.doi.org/10.5061/dryad.7hb5c59 | Dryad Digital Repository, 10.5061/dryad.7hb5c59 |
| Strickler SR, Powell AF, Mueller LA, Züst T, Jander G | 2019 | Rapid and independent evolution of ancestral and novel chemical defenses in a genus of toxic plants (Erysimum, Brassicaceae) | https://www.ncbi.nlm.nih.gov/bioproject/PRJNA563696/ | NCBI BioProject, PRJNA563696 |

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
