## [Decision Letter]

**Acceptance summary:**

In this work, the authors begin to look at the question of how new pathways evolve within the context of an intact organism already containing existing pathways. Specifically, they investigate how the evolution of a new defense metabolic pathway influences the diversity within a pre-existing defense metabolite within the *Erysiumum* plant genus.

**Decision letter after peer review:**

Thank you for submitting your article "Rapid and independent evolution of ancestral and novel defenses in a genus of toxic plants (*Erysimum*, Brassicaceae)" for consideration by *eLife*. Your article has been reviewed by three peer reviewers, and the evaluation has been overseen by a Reviewing Editor and Christian Hardtke as the Senior Editor. The following individuals involved in review of your submission have agreed to reveal their identity: Noah Whiteman (Reviewer #2); Matthew W. Hahn (Reviewer #3).

The reviewers have discussed the reviews with one another and the Reviewing Editor has drafted this decision to help you prepare a revised submission.

Summary:

The manuscript describes a genomic and evolutionary comparison of how de novo defense trait evolution can progress to diversity and how this may influence existing trait diversity.

Essential revisions:

There was uniform excitement about the data and topic. But there are significant concerns surrounding the phylogenetic analysis. These include, an easily correctable one for claiming enzyme orthology within the glucosinolate pathway and a more complicated set of comments on how to properly handle temporal and radiation analysis of evolution within *Erysimum* for the cardenolides. The reviewers have given explicit advice on these topics including suggested technical approaches to better handle this complexity and avenues where removing claims or caveating claims is more appropriate than new analysis. There was uniform excitement about the data and topic. But there are significant concerns surrounding the phylogenetic analysis. These include, an easily correctable one for claiming enzyme orthology within the glucosinolate pathway and a more complicated set of comments on how to properly handle temporal and radiation analysis of evolution within *Erysimum* for the cardenolides. The reviewers have given explicit advice on these topics including suggested technical approaches to better handle this complexity and avenues where removing claims or caveating claims is more appropriate than new analysis.

Please see the below reviews for more detailed discussions.

Reviewer #1:

In this manuscript, the authors conduct a very interesting assessment of *Erysimum* chemical variation starting with a foundation genome. Transcriptome work is used to aid in discerning phylogenetic relationships among the species to aid in the analysis. I'm presuming that the deeper use of the transcriptomes to look at gene family evolution is being reserved for follow-up manuscripts. Overall, this is a very strong manuscript and I largely only have points that require cleaner analysis and/or clarification. They will require some new data analysis but not experimentation.

I'm having a bit of difficulty in the annotation of glucosinolate biosynthetic genes in *E. cheiranthoides.* This analysis appears to be using a single BLAST score value to call homology without phylogenetic interpretation whereas in the myrosinase family, there is an excellent alignment based approach. There are a number of reasons that make me request that a similar phylogenetic approach be used for all steps in the biosynthetic pathway similar to what was done with myrosinase. First, the use of the threshold similarity value can be misleading if the divergence happened after the *Erysimum*/Arabidopsis split. For example, AOP2/AOP3 likely diverged within the Arabidopsis species and as such the difficulty of finding only AOP3 and not AOP2 is likely due to this fact. This is equally true for GS-OH where the similarity value would be similar between the non-functional and functional GS-OH enzymes in Arabidopsis. There are also very few species between *Erysimum* and Arabidopsis that have the GS-OH product or even precursor so it is not clear if we should expect conservation of the gene. Another indicator that phylogenetic approaches should be used is that a tree with just Arabidopsis species shows that there is not a 1:1 orthology conservation in the FMO GS-OX family and as such it is highly unlikely that this is true to *Erysimum*. I agree that for most steps, the data supports the presence of a gene, but to really clarify the relationships and predictions, this should be based on phylogenies for each gene family. Optimally, this would be using Brassica as an outgroup to help position the Arabidopsis/*Erysimum* relationships. This is true for both the aliphatic and indolic pathways.

The chemical diversity in glucosinolates is largely an epistatic interaction of the side-chain modifying genes and the MAM locus. In Table 2, principal components of the Bray-Curtiss Matrices are used to describe the chemical diversity. Given the heatmap in Figure 7, there is a tree relationship largely driven by the bottom cluster. Further, the authors are able to classify based on the ALK/MSO/MSI status. This suggests that the linear decomposition is creating a false negative. Another way to test this would be to create a genotypic matrix based on the imputed status of the major structural loci within each species and then compare the tree derived from this glucosinolate "genotype" to the genomic "Tree". Presently, I'm struggling to understand how Figure 7 provides such a "clear" image while Table 2 suggests that the chemotype/tree relationship is barely if at all there.

When the authors are discussing diversity of cardenolides vs. glucosinolates, it seems that the catabolic diversity of glucosinolates should be included. Cardenolides from my understanding function as intact molecules while each glucosinolate can be catabolized to at least three different structures (nitrile, thiocyanate or isothiocyanate) each with different activities. This suggests that the presence or absence of the ESP/TSP/NSP family should be investigated to assess if this might be making up for the lower glucosinolate diversity.

In both the cardenolide and glucosinolate pathway, is it proper to assume a linear gain of structural diversity? Both pathways appear to be structured to allow for non-linear gains in structural diversity. For example, in the glucosinolate pathway, the simple shift of 3C to 4C gains a suite of new compounds. Could the pathways structures be the reason that there is no evidence for a directional trend and instead a rapid gain? A discussion of how pathway shape influences the assumptions in the gain of chemical diversity should be included. Future work on using modelling to understand how pathway structure and enzyme "promiscuity" may be misleading our assumptions would be really cool but is way outside of the scope of this manuscript.

I'm somewhat confused by the description of the metabolite analysis. The Materials and methods section alludes to both pools and individuals being analyzed but it isn't clear how many samples were used per species for the figures. For example, in Figure 7, is there no observed within species genetic variation or are all the species selfing meaning that the samples were isogenic? Or are these averages that are blind to potential genetic variation? The same question is true for the cardenolides. Work by Cacho and Strauss showed extensive within species genetic variation across the *Streptanthus* for glucosinolates so it would be very helpful to clarify this.

Reviewer #2:

This is a comprehensive study on the origin and diversification of the remarkable *Erysimum* lineage within the Brassicaceae that contains species that produce both glucosinolates as well as cardenolides that deserves to be published in the current venue, in my mind. It is entirely “descriptive” and that's just fine. It provides a wonderful and systematic set of data that can leveraged by many different types of biologists and it provides a real example of the escape and radiate model of Ehrlich and Raven.

A major question that has evaded us in a general sense is whether there are tradeoffs in the production of these two classes of so-called secondary compounds, as one might expect given their metabolic costs. To test this, they needed to obtain a phylogeny, and the authors sequenced and assembled the genome of an exemplar species, *E. cheiranthoides* and obtained foliar transcriptomes for 47 additional *Erysimum* species. They also profiled the two chemical classes from these plant species. Further, they complemented this by obtaining glucosinolate-activating myrosinase activity data, inhibition of the animal target of the cardenolides (the sodium pump) by leaf extracts, and inducibility of these defenses in response to application of jasmonic acid, a standard, but not ideal (but often good enough) proxy for chewing herbivore (or necrotoph) attack. The functional work is laudable. They also propose a new pathway for the modification of the simplest cardenolide, digitoxigenin, which may aid in the identification of genes encoding enzymes predicted to be required at different points on the pathway.

The central questions that they address are: the relative timing of diversification of species in this group, and whether the diversity and amount of glucosinolates is correlated in any way with those in the cardenolides. They find rapid origin for most extant lineages in the clade, and they find a phylogenetic signal for cardenolides (closely related species are more similar in their profiles than less closely related species) but not glucosinolates.

My main feedback on the phylogenetics is that I thought some of the questions were under-explored. The authors should attempt to create a chronogram so that one could begin to place some dates on different diversification events and the origin of the homologies that are salient here. This will require more work and I know there just are not many fossils for calibration. But I think that the authors should try.

Cardenolides were thought to be a key innovation in the lineage *Erysimum* (linked to an escape and radiate model sensu Ehrlich and Raven, 1964) and this study actually tests this hypothesis. But it would be a bit more robust in my mind if the authors could explore the species diversification question a bit more. How much of the variation in diversification rate or speciation rate across *Erysimum* is explained by the origin of cardenolides, their diversity and abundance (and the same for glucosinolates)? The authors might be able to deploy a PGLS analysis and it would be fruitful to see the results of this.

Another question I had was whether there are guilds of herbivore species now adapted to both glucosinolates and cardenolides? I did see this possibility briefly mentioned but it would be great to have them elaborate a bit more. The reason is that this might help explain the evolution of the great diversity of cardenolide compounds seen in these *Erysimum* species.

There is a profundity of excellent natural history details and information that the field will appreciate on exceptions to the rule in this system (e.g., the species with no or very high cardenolides, the species that does induce cardenolide production in respond to JA).

Overall, I thought that this system helps address a fundamental question on the evolution of key new defenses in plants, and whether tradeoffs with ancestral compounds that are unrelated biochemically exist. They apparently don't, at least at first blush.

Reviewer #3:

This manuscript examines the variation in plant defensive metabolites across the genus *Erysimum*, presenting a well-argued motivation for why these should be studied. The manuscript contains a lot of different kinds of data, including a de novo genome assembly, metabolic profiles (under multiple conditions), measurements of enzyme activity, mass spectrometry, and transcriptome sequencing. The genome assembly looks good, and I know very little about the biochemical methods used here. As a result, my review focuses on the transcriptome sequencing and downstream phylogenetic analyses.

In short, the phylogenetics is lacking. The analyses presented are very incomplete, and some of them are wrong. That being said, there is a lot the authors could conclude without a phylogeny, including almost everything about the lack of correlation between glucosinolates and cardenolides. My comments therefore attempt to describe my concerns about the existing phylogenetic analyses and how they could be improved, and to point out where what is claimed in the current manuscript is unsupported by the data.

(I apologize ahead of time for the citations to my own work below. I do this not because I expect the papers to be cited, but because they are the easiest examples for me to recall. Feel free to cite alternatives.)

1) Phylogenomics in a radiation.

Perhaps the most interesting part of the phylogenetic data is undescribed: the huge amount of gene tree discordance. I can tell that there is a lot of discordance because the ASTRAL internal branches are extremely short, and these are inferred from the amount of discordance observed in the input gene trees. But the paper doesn't describe these gene trees at all, even though this sort of pattern (i.e. radiations with lots of incomplete lineage sorting and/or introgression) is one of the most interesting current patterns in phylogenomics. There are many recent plant examples of this, including Novikova et al., 2016; Pease et al., 2016; Copetti et al., 2017 and Wu et al., 2018.

This lack of results about discordance is not simply a subject that I think would be nice for the manuscript. Rather, I think these results are required to put the radiation in context, for a number of reasons. First, though the manuscript focuses on the statistical support for individual nodes, this support is misleading about the history of individual gene trees. One can have 100% statistical support for a branch, but still have most gene trees be discordant with it. So, at a basic descriptive level, I would argue that only reporting support values is misleading.

Second, as has been discussed in multiple recent papers about radiations and histories with discordance (e.g. Pease et al., 2016; Wu et al., 2018), standard phylogenetic methods looking at trait evolution will fail. This occurs because the traits are analyzed using the species tree, even though traits follow individual gene trees (this general problem has been termed "hemiplasy"). For example, diversity present in the common ancestor of some or all of the species in the genus may sort among species (via drift or selection), resulting in an incongruent trait pattern. But each of these traits only needed to arise once-there is no need to invoke complex evolutionary scenarios such as the one described.

How hemiplasy interacts with continuous traits is less clear and is an active area of research (e.g. Mendes et al., 2018). What is clear, however, is that simply simulating Brownian motion of a trait across a fixed species tree (as is done here in the analyses for Blomberg's K, for instance), is not a good model for how traits are truly evolving in clades such as this one.

In summary, there are a number of analyses that should be run to describe the data better (gene or site discordance), and a number of analyses and conclusions that should be re-thought in light of whatever levels of discordance are found.

2) Phylogenomic methods.

I had a number of basic questions about the analyses being done, and two suggested (though not necessary) analyses. First, the suggested analyses: why not date the *Erysimum* radiation here? Strangely, the authors rely on dating from a paper with one locus (I believe), when they should be able to do a much better job. Later in the manuscript the authors even dismiss these earlier papers because they used "conventional approaches" for phylogenetics (which I think means only a handful of loci), so it is strange to see this straightforward analysis lacking here. The second suggested analysis is of introgression: the authors have more than enough data to test for introgression, either across the tree as a whole or in individual parts (e.g. using the ABBA-BABA test). The manuscript vaguely invokes the possibility of introgression in this genus, but why not test for it directly?

In terms of the analyses that are done, I noticed that the ASTRAL tree presented has equal branch lengths across all tips. Because ASTRAL infers branch lengths from levels of discordance, it cannot directly estimate tip branches when one sample per species is used. It appears that the authors have simply used a standard unit branch length for all the tips (in my experience, ASTRAL assigns these branches a length of 1 coalescent unit, but no units are given on this tree, so it is hard to tell). This usage is incorrect, as the ASTRAL tree has no information about tip branches. The authors should either present a different tree or find a different way to express terminal branch lengths.

The authors do seem to use branch lengths in substitutions per site when they need them for the comparative methods, but these lengths come from a different dataset (with one-quarter as many loci) and a tree with a different topology. Instead, if the authors want to use this type of branch length, they should either make a concatenated tree from the full dataset or use a method like StarBeast2 (Ogilvie et al., 2017, MBE) to infer proper branch lengths.

I also wondered why there is a whole other method for identifying orthologs presented. This method is both very confusing (i.e. I could not follow the methods) and it is basically not used. The only place it really seems to be used is in constructing a tree with branch lengths in substitutions per site, but it would be much more appropriate to do this with the larger dataset (and would allow for a direct comparison of concatenation and ASTRAL using the same dataset; plus, the branch to the species with a genome is strangely long in the tree that is presented). Furthermore, comparison of the trees reveals a bunch of differences, including in branches supported by posterior probabilities of 1, unlike what is stated in the main text. Maybe drop this second dataset?

3) Phylogenetic comparative methods and ancestral states.

As alluded to earlier, I think there are real problems with applying standard phylogenetic comparative methods to radiations. This affects not only the inferred modes of evolution, but the rates as well. Given that the main effect of discordance should be in erasing phylogenetic signal, it was actually surprising to me that so many traits had significant values of Blomberg's K that were >1. I think this result means that lots of similarities are driven by geographical area, so that variance between clades (which are geographically structured) is much greater than variance within clades. It's also interesting that other traits do not show these patterns, and again I want to point out that similar differences between traits have been found in other radiations (e.g. Wu et al., 2018).

More importantly, I think there are two problems in how the trees and analyses are interpreted:

First, the authors claim that there is rapid evolution of cardenolides. It is not clear where this comes from. The only test I could find is for directionality of cardenolides, as opposed to a Brownian motion model. I am unsure how this is a test for rapid evolution (especially since it is the lack of significance that appears to be taken as positive evidence), nor how any trait would pass such a test on the tree being used. If the rapid evolution of these traits is going to be a major conclusion of the current manuscript (it's in the title, after all), a much clearer description of the evidence for this is needed.

Second, throughout the manuscript the authors refer to "early-diverging" species as if they somehow represent the ancestral state. Phylogenetic trees cannot be read in this manner, and there is nothing more ancestral about species on one side or the other of the deepest splits in a tree. All of this language and conclusions should be fixed.

[Editors' note: further revisions were suggested prior to acceptance, as described below.]

Thank you for submitting your article "Independent evolution of ancestral and novel defenses in a genus of toxic plants (*Erysimum*, Brassicaceae)" for consideration by *eLife*. Your article has been reviewed by three peer reviewers, and the evaluation has been overseen by a Reviewing Editor and Christian Hardtke as the Senior Editor. The following individuals involved in review of your submission have agreed to reveal their identity: Noah Whiteman (Reviewer #2); Matthew W. Hahn (Reviewer #3).

The reviewers have discussed the reviews with one another and the Reviewing Editor has drafted this decision to help you prepare a revised submission.

Summary:

In this work, the authors test how the evolution of a new defense metabolite influences or alters the trajectory of evolution in a pre-existing defense metabolite.

Essential revisions:

There are three key factors that need to be addressed.

1) The strength of claims around the concordance factors and general wording needs correcting.

2) Multiple testing control needs to be described.

3) A description of if these species fitting the model assumptions.

Reviewer #3:

In this revised manuscript, the authors have done an admirable job updating both their analyses and their conclusions. Even for the analyses still included that I don't quite agree with, I think the authors have done well in explaining what the problems might be. I thank them for all their hard work.

Given this, I do have several suggestions about how to clarify the new analyses and the way they are discussed.

1) The language surrounding concordance factors (and discordance in general) has a few problems.

a) For instance, the authors say: "Despite considerable discordance in the ASTRAL species tree…" But this is not right: the discordance is in the species tree, not somehow only in the ASTRAL tree. The same short branches are obvious in the ExaML tree, and there is also discordance with it. This problem would most helpfully be solved by displaying the concordance factors on the tree shown in the main text (regardless of which algorithm produced it) – this is easily done with the numbers the authors have already calculated.

b) The "distinct geographic clades" referred to in the text all seem to be supported by equal to or less than ~50% of gene trees. While this does not make them any less correlated with geography, it also doesn't mean that gene tree discordance across these clades should be ignored. (The agreement between the ASTRAL topology and the ExaML topology on these clades is not extra evidence of anything.)

c) I think the text misstates what the concordance factors are: these are not only a proportion relative to the top three topologies. These are the proportion of all informative trees that support the most common topology (though ASTRAL will also report the proportion supporting the second- and third-most common topologies).

2) I had a question about the detected phylogenetic signal in cardenolides. I noticed that the text reports a "strong correlation" between cardenolide profile and phylogenetic relatedness, but when I tried to compare the cardenolide chemogram to the phylogeny (by eye), I found that they were very different, except for one "clade" of Spanish/Moroccan plants. How can these two things both be true?

Similarly, can the authors explain the two apparently contradictory patterns reported in this sentence: "In contrast, species formed less distinct clusters based on similarity in cardenolide defenses, but more related species shared more similar cardenolide profiles".

Is there a biological interpretation to this?

3) The introgression analyses are not very clear. First, HyDE is explicitly testing for homoploid hybrid speciation. Is this what the authors think is happening in *Erysimum*? I would have thought that a standard D-test would be more agnostic about patterns of discordance due to introgression.

Second, I can't find a list of results or a description of how many tests were carried out, etc. for these tests. The supplementary figure simply says that significant results are shown.

4) The authors have added a bunch of new analyses of speciation rates but have failed to correct for the many tests they have carried out. These analyses calculate "tip-specific" speciation rates in two different ways, but then test for associations with multiple measures of both glucosinolates and cardenolides, and in multiple different ways.

Multiple-test corrections must be carried out, including correcting for tests on variables that are reported as "not shown." The discussion of these results should also be updated to reflect corrected p-values.

[Editors' note: further revisions were suggested prior to acceptance, as described below.]

Thank you for resubmitting your work entitled "Independent evolution of ancestral and novel defenses in a genus of toxic plants (*Erysimum*, Brassicaceae)" for further consideration by *eLife*. Your revised article has been evaluated by Christian Hardtke (Senior Editor) and a Reviewing Editor.

The manuscript has been improved but there is one tweak suggested to clarify one aspect of the Discussion.

Reviewer #3:

I thank the authors again for all their changes and clarifications. The text is now much clearer, and I think the methods are described in better detail (I do apologize for having missed the fact that they had previously used quartet scores and not concordance factors).

My one remaining comment involves the new text describing all of the polyploid taxa in this clade, and the possibility that orthologs have been misidentified, leading to higher-than-expected discordance. While I think it is good to raise this possibility here, I request that the authors add a note of caution to the interpretation of the hybridization analyses, as these could also be affected by the same problem (e.g. allopolyploidy could look like between-species introgression).

---

## [Author Response]

Reviewer #1:[…]I'm having a bit of difficulty in the annotation of glucosinolate biosynthetic genes in E. cheiranthoides. This analysis appears to be using a single BLAST score value to call homology without phylogenetic interpretation whereas in the myrosinase family, there is an excellent alignment based approach. There are a number of reasons that make me request that a similar phylogenetic approach be used for all steps in the biosynthetic pathway similar to what was done with myrosinase. First, the use of the threshold similarity value can be misleading if the divergence happened after the Erysimum/Arabidopsis split. For example, AOP2/AOP3 likely diverged within the Arabidopsis species and as such the difficulty of finding only AOP3 and not AOP2 is likely due to this fact. This is equally true for GS-OH where the similarity value would be similar between the non-functional and functional GS-OH enzymes in Arabidopsis. There are also very few species between Erysimum and Arabidopsis that have the GS-OH product or even precursor so it is not clear if we should expect conservation of the gene. Another indicator that phylogenetic approaches should be used is that a tree with just Arabidopsis species shows that there is not a 1:1 orthology conservation in the FMO GS-OX family and as such it is highly unlikely that this is true to Erysimum. I agree that for most steps, the data supports the presence of a gene, but to really clarify the relationships and predictions, this should be based on phylogenies for each gene family. Optimally, this would be using Brassica as an outgroup to help position the Arabidopsis/Erysimum relationships. This is true for both the aliphatic and indolic pathways.

We have constructed and added phylogenetic trees of aliphatic and indole glucosinolate genes using sequences from *E. cheiranthoides, A. thaliana*, and *B. oleracea.* These have been added to the revised manuscript as figure supplements to Figures 3 and 4. Note that some *E. cheiranthoides* gene names have changed due to further refinement of the genome assembly and annotation since the previous submission of this manuscript.

The chemical diversity in glucosinolates is largely an epistatic interaction of the side-chain modifying genes and the MAM locus. In Table 2, principal components of the Bray-Curtiss Matrices are used to describe the chemical diversity. Given the heatmap in Figure 7, there is a tree relationship largely driven by the bottom cluster. Further, the authors are able to classify based on the ALK/MSO/MSI status. This suggests that the linear decomposition is creating a false negative. Another way to test this would be to create a genotypic matrix based on the imputed status of the major structural loci within each species and then compare the tree derived from this glucosinolate "genotype" to the genomic "Tree". Presently, I'm struggling to understand how Figure 7 provides such a "clear" image while Table 2 suggests that the chemotype/tree relationship is barely if at all there.

The heatmap in Figure 7 is structured by chemical similarity, and the “tree” in panel A is a chemogram that clusters together the chemically most similar species. The very clear structure in this chemogram (and the lack thereof in Figure 10) indicates that glucosinolate chemotypes are likely driven by a small number of major effect genes (mainly the MAM and ALK/MSO/MSI loci), whereas cardenolide chemotypes appear to be driven by more minor effect genes with less distinct distributions among species. However, there is no direct link of this figure to phylogenetic relatedness, explaining why the first and second coordinates of the glucosinolate PCoA show no phylogenetic signal. We have now added two analyses to clarify these results. First, we have added Mantel tests that compare the full chemical dissimilarity matrix to the cophenetic distance matrix, with equivalent results. In addition, we have added a figure supplement (Figure 5—figure supplement 3) in which we reconstruct ancestral states for total glucosinolate/cardenolide compounds and for the first principal coordinates of each compound dissimilarity matrix. This highlights that the lack of phylogenetic signal for glucosinolates is driven by several clades in which closely related species evolved in opposite directions on the first PCoA.

When the authors are discussing diversity of cardenolides vs. glucosinolates, it seems that the catabolic diversity of glucosinolates should be included. Cardenolides from my understanding function as intact molecules while each glucosinolate can be catabolized to at least three different structures (nitrile, thiocyanate or isothiocyanate) each with different activities. This suggests that the presence or absence of the ESP/TSP/NSP family should be investigated to assess if this might be making up for the lower glucosinolate diversity.

This is an excellent point, and we have included it as part of our discussion. Unfortunately, we did not measure glucosinolate breakdown products on the 48 species, which would be the most direct test of potential “hidden” chemical diversity. We have, however, constructed a phylogenetic tree of the ESP/TSP/NSP family from *E. cheiranthoides, A. thaliana*, and *B. oleracea* (Figure 4—figure supplement 11). This shows the likely presence of these genes in *E. cheiranthoides* and suggests that there is variation in glucosinolate breakdown.

In both the cardenolide and glucosinolate pathway, is it proper to assume a linear gain of structural diversity? Both pathways appear to be structured to allow for non-linear gains in structural diversity. For example, in the glucosinolate pathway, the simple shift of 3C to 4C gains a suite of new compounds. Could the pathways structures be the reason that there is no evidence for a directional trend and instead a rapid gain? A discussion of how pathway shape influences the assumptions in the gain of chemical diversity should be included. Future work on using modelling to understand how pathway structure and enzyme "promiscuity" may be misleading our assumptions would be really cool but is way outside of the scope of this manuscript.

We agree that biosynthesis of both compounds should be considered as networks rather than linear pathways, and future analysis of pathway structures would be a highly exciting area of research. For the current study we maintain the more linear presentation of biosynthesis but highlight that this likely is an oversimplification.

I'm somewhat confused by the description of the metabolite analysis. The Materials and methods section alludes to both pools and individuals being analyzed but it isn't clear how many samples were used per species for the figures. For example, in Figure 7, is there no observed within species genetic variation or are all the species selfing meaning that the samples were isogenic? Or are these averages that are blind to potential genetic variation? The same question is true for the cardenolides. Work by Cacho and Strauss showed extensive within species genetic variation across the Streptanthus for glucosinolates so it would be very helpful to clarify this.

None of the plants used for chemical analysis were isogenic (*E. cheiranthoides* used for chemistry was F1), but nonetheless genetic variation was likely quite low, with seeds mostly originating from single mothers. In addition, species obtained from Botanical Gardens likely experienced significant inbreeding, as they often had been cultivated in small artificial populations for several years after initial collection from the field. Due to the low availability of multiple natural accessions for most species and varying degrees of “natural” levels of genetic diversity among species, we decided to use an averaging approach that is intentionally blind to intraspecific variation. We added statements to make this more explicit. However, at least for the annual *E. cheiranthoides* we have evidence that intraspecific variation is lower than might be expected. Among a collection of more than 20 natural accessions we find no variation in glucosinolate profiles, and only moderate variation in cardenolide profiles.

Reviewer #2:[…]My main feedback on the phylogenetics is that I thought some of the questions were under-explored. The authors should attempt to create a chronogram so that one could begin to place some dates on different diversification events and the origin of the homologies that are salient here. This will require more work and I know there just are not many fossils for calibration. But I think that the authors should try.

We have added a new phylogenetic analysis in which we use ExaML to generate a concatenated phylogeny with branch length information for the full 9,868 genes, and then use treePL to generate an ultrametric chronogram for the 48 species. See Materials and methods and response to reviewer #3 for additional information.

Cardenolides were thought to be a key innovation in the lineage Erysimum (linked to an escape and radiate model sensu Ehrlich and Raven, 1964) and this study actually tests this hypothesis. But it would be a bit more robust in my mind if the authors could explore the species diversification question a bit more. How much of the variation in diversification rate or speciation rate across Erysimum is explained by the origin of cardenolides, their diversity and abundance (and the same for glucosinolates)? The authors might be able to deploy a PGLS analysis and it would be fruitful to see the results of this.

We have added tip-rate correlation analyses to test if variation in glucosinolate and cardenolide compounds is correlated with tip-specific speciation rates across the phylogeny (estimated as node densities, ND, and inverse equal split, ES). We test for these correlations using PGLS models, as well as a recently proposed, robust simulation-based method. The different speciation rates measures and statistical methods largely agree in their main results and reveal a lack of correlation between glucosinolate compound numbers or concentrations and speciation rates. In contrast, there is a positive correlation between the number of cardenolide compounds and speciation rate that is significant or marginally significant depending on speciation rate metric, with variation in cardenolide compound number explaining between 17-28% of variation in speciation rates.

For consistency we now also use PGLS analyses to test for phylogenetically corrected correlations among traits. While results do not change and traits remain uncorrelated, we believe this makes an even stronger argument for an apparent lack of trade-offs between glucosinolates and cardenolides.

Another question I had was whether there are guilds of herbivore species now adapted to both glucosinolates and cardenolides? I did see this possibility briefly mentioned but it would be great to have them elaborate a bit more. The reason is that this might help explain the evolution of the great diversity of cardenolide compounds seen in these Erysimum species.

This is an excellent point, and while our main point holds true that *Erysimum* herbivores have evolved cardenolide resistance de novo, it seems likely that cardenolides may facilitate host shifts in cardenolide-specialist herbivores. Interestingly, the reviewer’s observation is matched by an observation of our co-author Georg Petschenka, who observed a cardenolide-resistant milkweed bug that normally feeds on *Digitalis* has shifted to *Erysimum crepidifolium* where this food source is available. We now provide this example as part of our Introduction.

There is a profundity of excellent natural history details and information that the field will appreciate on exceptions to the rule in this system (e.g., the species with no or very high cardenolides, the species that does induce cardenolide production in respond to JA).Overall, I thought that this system helps address a fundamental question on the evolution of key new defenses in plants, and whether tradeoffs with ancestral compounds that are unrelated biochemically exist. They apparently don't, at least at first blush.

We thank the reviewer for his kind words. We find these results exciting and have further emphasized the lack of trade-offs and apparent independent evolution of glucosinolates and cardenolides as the main result of this study.

Reviewer #3:[…]1) Phylogenomics in a radiation.Perhaps the most interesting part of the phylogenetic data is undescribed: the huge amount of gene tree discordance. I can tell that there is a lot of discordance because the ASTRAL internal branches are extremely short, and these are inferred from the amount of discordance observed in the input gene trees. But the paper doesn't describe these gene trees at all, even though this sort of pattern (i.e. radiations with lots of incomplete lineage sorting and/or introgression) is one of the most interesting current patterns in phylogenomics. There are many recent plant examples of this, including Novikova et al., 2016; Pease et al., 2016; Copetti et al., 2017 and Wu et al., 2018.This lack of results about discordance is not simply a subject that I think would be nice for the manuscript. Rather, I think these results are required to put the radiation in context, for a number of reasons. First, though the manuscript focuses on the statistical support for individual nodes, this support is misleading about the history of individual gene trees. One can have 100% statistical support for a branch, but still have most gene trees be discordant with it. So, at a basic descriptive level, I would argue that only reporting support values is misleading.

We thank the reviewer for highlighting these issues. We have now significantly modified our phylogenomic analyses. Focusing on the main dataset of 9,868 orthologous genes (and dropping the second set of trees using alternate methods, see below), we now present an ASTRAL tree with added discordance vales for each node (Figure 5—figure supplement 1, Supplementary file 4). As already predicted by the reviewer from the short internal branch lengths on the previous ASTRAL tree, we found significant levels of discordance, likely driven by the high levels of hybridization among species (see below).

We also generated a concatenated tree using ExaML, as this method could better handle the computational challenge of working with a set of 9,868 genes. Although there are considerable levels of discordance between the gene trees, we still recover the major clades as congruent. We believe this tree to provide meaningful insights, namely a strong signal of geographic area in phylogenetic relatedness that appears to be driving the chemical phenotypes to different degrees.

Second, as has been discussed in multiple recent papers about radiations and histories with discordance (e.g. Pease et al., 2016; Wu et al., 2018), standard phylogenetic methods looking at trait evolution will fail. This occurs because the traits are analyzed using the species tree, even though traits follow individual gene trees (this general problem has been termed "hemiplasy"). For example, diversity present in the common ancestor of some or all of the species in the genus may sort among species (via drift or selection), resulting in an incongruent trait pattern. But each of these traits only needed to arise once-there is no need to invoke complex evolutionary scenarios such as the one described.

We have removed these arguments and instead added discussion points of how hemiplasy could affect the interpretation of our results.

How hemiplasy interacts with continuous traits is less clear and is an active area of research (e.g. Mendes et al., 2018). What is clear, however, is that simply simulating Brownian motion of a trait across a fixed species tree (as is done here in the analyses for Blomberg's K, for instance), is not a good model for how traits are truly evolving in clades such as this one.

We acknowledge the limitations of analyses based on a single fixed species tree and agree that a more sophisticated analyses of trait evolution would be required to understand the true evolutionary histories in this complex species radiation. Nonetheless, we argue that for a broadly comparative study such as ours, a simple analysis such as Blomberg’s K still provides valuable first insights.

In summary, there are a number of analyses that should be run to describe the data better (gene or site discordance), and a number of analyses and conclusions that should be re-thought in light of whatever levels of discordance are found.

We hope that our modified analyses and discussion of results now appropriately reflect the levels of discordance found within our species tree.

2) Phylogenomic methods.I had a number of basic questions about the analyses being done, and two suggested (though not necessary) analyses. First, the suggested analyses: why not date the Erysimum radiation here? Strangely, the authors rely on dating from a paper with one locus (I believe), when they should be able to do a much better job. Later in the manuscript the authors even dismiss these earlier papers because they used "conventional approaches" for phylogenetics (which I think means only a handful of loci), so it is strange to see this straightforward analysis lacking here. The second suggested analysis is of introgression: the authors have more than enough data to test for introgression, either across the tree as a whole or in individual parts (e.g. using the ABBA-BABA test). The manuscript vaguely invokes the possibility of introgression in this genus, but why not test for it directly?

We now have included a dated, ultrametric ExaML tree which provides estimates of divergence time. This analysis used treePL in which we relied on a few estimates from published studies to constrain specific nodes (see Materials and methods). Without these constraints, divergence time estimates would not have matched the general consensus of a very recent radiation of the genus. However, it seems feasible that polyploidization, chimerism, and hybridization could result in incorrect estimation of divergence times in our dataset. As such, we acknowledge the relatively weak confidence in our own divergence time estimates, and we also removed all statements dismissing earlier papers’ estimates.

To quantify hybridization we used HyDe (see Materials and methods), which revealed high levels of hybridization across all 48 species and provides a likely explanation for the high levels of discordance.

In terms of the analyses that are done, I noticed that the ASTRAL tree presented has equal branch lengths across all tips. Because ASTRAL infers branch lengths from levels of discordance, it cannot directly estimate tip branches when one sample per species is used. It appears that the authors have simply used a standard unit branch length for all the tips (in my experience, ASTRAL assigns these branches a length of 1 coalescent unit, but no units are given on this tree, so it is hard to tell). This usage is incorrect, as the ASTRAL tree has no information about tip branches. The authors should either present a different tree or find a different way to express terminal branch lengths.

We now present an ASTRAL tree with terminal branch length manually adjusted to 0.2 coalescent units as a supplementary figure to highlight tree discordance (Figure 5—figure supplement 1). As our main tree we present a concatenated and dated ExaML tree in which branch lengths correspond to estimated divergence times.

The authors do seem to use branch lengths in substitutions per site when they need them for the comparative methods, but these lengths come from a different dataset (with one-quarter as many loci) and a tree with a different topology. Instead, if the authors want to use this type of branch length, they should either make a concatenated tree from the full dataset or use a method like StarBeast2 (Ogilvie et al., 2017, MBE) to infer proper branch lengths.

While we have removed the main inadequate analysis that relied on branch lengths as substitutions per site (see below), we now use an ultrametric ExaML tree for all our remaining and new analyses. There are several differences in topology between the ASTRAL and ExaML trees, likely reflecting the high degree of gene discordance among species. However, the main geographic clades are conserved between the trees, and the direction of results from trait analyses remained unchanged, thus we believe this approach is adequate for an evaluation of general patterns in our chemistry data.

I also wondered why there is a whole other method for identifying orthologs presented. This method is both very confusing (i.e. I could not follow the methods) and it is basically not used. The only place it really seems to be used is in constructing a tree with branch lengths in substitutions per site, but it would be much more appropriate to do this with the larger dataset (and would allow for a direct comparison of concatenation and ASTRAL using the same dataset; plus, the branch to the species with a genome is strangely long in the tree that is presented). Furthermore, comparison of the trees reveals a bunch of differences, including in branches supported by posterior probabilities of 1, unlike what is stated in the main text. Maybe drop this second dataset?

Following the reviewer’s suggestion we removed the separate method for identifying orthologs and tree construction.

3) Phylogenetic comparative methods and ancestral states.As alluded to earlier, I think there are real problems with applying standard phylogenetic comparative methods to radiations. This affects not only the inferred modes of evolution, but the rates as well. Given that the main effect of discordance should be in erasing phylogenetic signal, it was actually surprising to me that so many traits had significant values of Blomberg's K that were >1. I think this result means that lots of similarities are driven by geographical area, so that variance between clades (which are geographically structured) is much greater than variance within clades. It's also interesting that other traits do not show these patterns, and again I want to point out that similar differences between traits have been found in other radiations (e.g. Wu et al., 2018).

We acknowledge the limitation of standard phylogenetic analyses using single fixed trees and have added discussion points throughout to highlight these limitations and place our results better in the context of other recent studies. However, we decided to keep the analyses of Blomberg’s K as we believe they still provide meaningful results. Phylogenetic distances captured by both the ExaML and ASTRAL trees are highly correlated with geographic distances, and as such we agree that geographical area is likely a major driver of similarities in cardenolides, as captured by Blomberg’s K. Nonetheless, the lack of an equivalent signal in glucosinolates indicates that different mechanisms are determining glucosinolate and cardenolide phenotypes. To visualize these differences even better, we also added new analyses in which we reconstruct the ancestral states of total concentrations and the first PCoA for both glucosinolates and cardenolides (Figure 5—figure supplement 3). Together, these results clearly demonstrate that closely related and co-occurring species share the same cardenolide but distinct glucosinolate phenotypes.

More importantly, I think there are two problems in how the trees and analyses are interpreted:First, the authors claim that there is rapid evolution of cardenolides. It is not clear where this comes from. The only test I could find is for directionality of cardenolides, as opposed to a Brownian motion model. I am unsure how this is a test for rapid evolution (especially since it is the lack of significance that appears to be taken as positive evidence), nor how any trait would pass such a test on the tree being used. If the rapid evolution of these traits is going to be a major conclusion of the current manuscript (it's in the title, after all), a much clearer description of the evidence for this is needed.

We have removed all mentions of speed from the text and title and have removed the test of directionality. We generally de-emphasize the speed of trait changes, but rather focus on the apparently independent evolution of glucosinolate and cardenolide traits.

Second, throughout the manuscript the authors refer to "early-diverging" species as if they somehow represent the ancestral state. Phylogenetic trees cannot be read in this manner, and there is nothing more ancestral about species on one side or the other of the deepest splits in a tree. All of this language and conclusions should be fixed.

We have carefully checked the text and removed all mentions of “early-diverging”.

[Editors' note: further revisions were suggested prior to acceptance, as described below.]

Essential revisions:There are three key factors that need to be addressed.1) The strength of claims around the concordance factors and general wording needs correcting.2) Multiple testing control needs to be described.3) A description of if these species fitting the model assumptions.

In response to the three essential revision requests, we have implemented concordance factors on our phylogeny (previously another metric of discordance was used) and made significant edits to the manuscript text to reflect the significant amounts of uncertainty in our phylogenetic analysis. We have modified the section criticized for multiple tests and provide evidence that false positives are unlikely to affect these results, and we argue that the HyDe test for hybridization is robust in response to unknown are likely multiple types of speciation.

Reviewer #3:[…]1) The language surrounding concordance factors (and discordance in general) has a few problems.a) For instance, the authors say: "Despite considerable discordance in the ASTRAL species tree…" But this is not right: the discordance is in the species tree, not somehow only in the ASTRAL tree. The same short branches are obvious in the ExaML tree, and there is also discordance with it. This problem would most helpfully be solved by displaying the concordance factors on the tree shown in the main text (regardless of which algorithm produced it) – this is easily done with the numbers the authors have already calculated.

We are aware that discordance is an inherent problem of the species tree and not just the ASTRAL tree. Our point here was simply that because there is this high level of discordance, it is unsurprising that the main topologies of the ExaML and ASTRAL trees don’t fully agree. For the same reason, we cannot use the numbers calculated for the ASTRAL tree to display discordance of the ExaML tree (note that values on the ASTRAL tree are quartet support values, not concordance factors). Instead, we have now calculated gene concordance factors (gCF) for the ExaML tree and display these on the main tree figure.

b) The "distinct geographic clades" referred to in the text all seem to be supported by equal to or less than ~50% of gene trees. While this does not make them any less correlated with geography, it also doesn't mean that gene tree discordance across these clades should be ignored. (The agreement between the ASTRAL topology and the ExaML topology on these clades is not extra evidence of anything.)

Most of the geographic clades of the main species tree are in fact supported by less than 5% of the gene trees when evaluated as concordance factors and have equivalent support over the first and second alternative when evaluated as quartet support values. We have further revised our manuscript to provide a more comprehensive discussion of the discordance results and highlight the combination of factors likely responsible, including methodological issues that could have inflated measures of discordance. We also shortened the discussion of geographic clades and emphasize that there is very low gene concordance supporting these clades. Nonetheless, we argue that the geographic clades uncovered by the main topology of our species tree are biologically meaningful given the high rate of endemism and often highly restricted geographic ranges among species. We maintain the set of phylogenetic analyses that uses the main species tree, and we are convinced that the main results from these analyses are robust even if the species tree is unlikely to be a perfect representation of species relatedness.

c) I think the text misstates what the concordance factors are: these are not only a proportion relative to the top three topologies. These are the proportion of all informative trees that support the most common topology (though ASTRAL will also report the proportion supporting the second- and third-most common topologies).

The estimates of discordance displayed in the ASTRAL tree are not concordance factors but quartet support values for the main topology (q1) relative to the first and second alternative. The minimum of q1 values is therefore of 33%, corresponding to complete discordance. We now have added concordance factors to the ExaML tree, and these are indeed the proportion of all informative trees that support the most common topology. For most deeper nodes, concordance factors are around 1%, highlighting very high levels of gene discordance.

2) I had a question about the detected phylogenetic signal in cardenolides. I noticed that the text reports a "strong correlation" between cardenolide profile and phylogenetic relatedness, but when I tried to compare the cardenolide chemogram to the phylogeny (by eye), I found that they were very different, except for one "clade" of Spanish/Moroccan plants. How can these two things both be true?

There is indeed a strong statistical correlation between phylogenetic relatedness and similarity in cardenolide profiles. However, this correlation is mostly not driven by whole clades, and therefore it is not easy to see by eye. To improve visualization, we have added co-phylogenetic plots as supplementary figures that optimize tip-to-tip matches between the ExaML phylogeny and the glucosinolate or cardenolide chemograms (Figure 5—figure supplement 4). Two distinct patterns are apparent from these figures that support a phylogenetic signal for cardenolide similarity. First, the number of species pairs for which the closest phylogenetic neighbor is also the most chemically similar species is higher for cardenolides than glucosinolates (12 vs 5 species pairs). Second, the total length of tip-to-tip links between phylogeny and chemograms is approximately twice as high for glucosinolates than it is for cardenolides (quantified by the minRotate coefficients of *cophylo*), which is clearly visible from many more long links. This highlights how closely related species are more often chemically distant for glucosinolates than for cardenolides.

Similarly, can the authors explain the two apparently contradictory patterns reported in this sentence: "In contrast, species formed less distinct clusters based on similarity in cardenolide defenses, but more related species shared more similar cardenolide profiles".Is there a biological interpretation to this?

“Species forming less distinct clusters” referred to the grouping of species into distinct chemotypes. For cardenolides, species could not be grouped into obviously distinct chemotypes and variation in profiles was less often due to clear presence/absence of compounds. The biological interpretation is that glucosinolate chemotypes are likely controlled by a small number of major-effect genes, while cardenolide diversity is more likely generated by many minor-effect genes. We have modified the discussion of these patterns to make this point clearer.

3) The introgression analyses are not very clear. First, HyDE is explicitly testing for homoploid hybrid speciation. Is this what the authors think is happening in Erysimum? I would have thought that a standard D-test would be more agnostic about patterns of discordance due to introgression.

The mode of speciation for each species of *Erysimum* is not known, but we believe that HyDe, which allows us to test for patterns of hybridization across the tree at once, provides evidence for the conclusions we make. If multiple modes of speciation are being utilized by *Erysimum*, we may be underestimating hybridization, but since we already are estimating widespread hybridization, we do not think this is a problem.

Second, I can't find a list of results or a description of how many tests were carried out, etc. for these tests. The supplementary figure simply says that significant results are shown.

HyDe ran a total of 51,888 tests, from which the software automatically selects the subset of significant results using a Bonferroni corrected p-value. We have added these details to the figure legend and also provide the full list of tests as a file on the Dryad repository.

4) The authors have added a bunch of new analyses of speciation rates but have failed to correct for the many tests they have carried out. These analyses calculate "tip-specific" speciation rates in two different ways, but then test for associations with multiple measures of both glucosinolates and cardenolides, and in multiple different ways.Multiple-test corrections must be carried out, including correcting for tests on variables that are reported as "not shown." The discussion of these results should also be updated to reflect corrected p-values.

Tests for correlations with speciation rates were added in response to a request from reviewer #2 in the previous revision. The results from these tests suggest that the number of cardenolide compounds and tip-specific speciation rates correlate positively, which we believe is an interesting finding. However, we are well aware of the limitations of this approach, particularly given that all analyses rely on the single species tree, which we know to suffer from high level of gene discordance. In consequence, these findings are neither a central aspect of our results nor of our discussion.

While we acknowledge these limitations, we disagree that multiple-test correction “must be carried out” to make these results more valid. It is obvious that any crude multiple-test correction such as Bonferroni-correction would make modestly significant correlations, such as we found them between cardenolide numbers and speciation rate, become non-significant. However, it is also well known that such corrections are excessively conservative and inflate Type II error.

We test for correlations in just four traits, the number and concentrations of cardenolides and glucosinolates. Instead of testing a bunch of different analyses and only presenting one test with significant results, we present four tests for each trait, since it is our impression that there is no clear consensus on how to best test for correlations with speciation rates. This brings the total number of tests to 16, but this would still mean we would expect less than one test to be false positive at p=0.05. Furthermore, the four tests per trait are hardly independent: the two speciation rate metrics are highly correlated and only differ slightly in relative weighting of diversification history, and PGLS and simulation tests attempt to model the same correlation but have different methodological biases. Given that three out of four tests involving a single trait are significant or marginal, this strengthens rather than weakens our confidence that this trait may be correlated with speciation rate.

We have added the missing non-significant test results and now present all 16 tests as the new Table 3. Instead of performing multiple test correction, we have performed a simulation test in which we generated random trait values from a normal distribution, performed all four tests on these random traits, and evaluated how often at least two of the tests were significant at the p=0.05 level. For a set of 1000 random traits, just three met these criteria, thus we conclude that false positives were highly unlikely to affect our results.

[Editors' note: further revisions were suggested prior to acceptance, as described below.]

Reviewer #3:I thank the authors again for all their changes and clarifications. The text is now much clearer, and I think the methods are described in better detail (I do apologize for having missed the fact that they had previously used quartet scores and not concordance factors).My one remaining comment involves the new text describing all of the polyploid taxa in this clade, and the possibility that orthologs have been misidentified, leading to higher-than-expected discordance. While I think it is good to raise this possibility here, I request that the authors add a note of caution to the interpretation of the hybridization analyses, as these could also be affected by the same problem (e.g. allopolyploidy could look like between-species introgression).

We have added the requested statement to our Discussion.